# Genetic diversity fuels gene discovery for tobacco and alcohol use

Tobacco and alcohol use are heritable behaviours associated with 15% and 5.3% of worldwide deaths, respectively, due largely to broad increased risk for disease and injury[1–4]. These substances are used across the globe, yet genome-wide association studies have focused largely on individuals of European ancestries[5]. Here we leveraged global genetic diversity across 3.4 million individuals from four major clines of global ancestry (approximately 21% non-European) to power the discovery and fine-mapping of genomic loci associated with tobacco and alcohol use, to inform function of these loci via ancestry-aware transcriptome-wide association studies, and to evaluate the genetic architecture and predictive power of polygenic risk within and across populations. We found that increases in sample size and genetic diversity improved locus identification and fine-mapping resolution, and that a large majority of the 3,823 associated variants (from 2,143 loci) showed consistent effect sizes across ancestry dimensions. However, polygenic risk scores developed in one ancestry performed poorly in others, highlighting the continued need to increase sample sizes of diverse ancestries to realize any potential benefit of polygenic prediction.

We developed a multi-ancestry meta-regression method to meta-analyse ancestrally diverse genome-wide association study (GWAS) summary statistics from 60 cohorts with 3,383,199 individuals (Supplementary Table 1; see Supplementary Fig. 1 for an overview of the project), representing major clines of recent human ancestry (Fig. 1a). The meta-analytic method used here uses meta-regression to account for per study axes of genetic ancestry variation combined with a random effect to capture further unexplained heterogeneity in the effect of a given genetic variant. Although ancestry here is continuous, we also performed secondary analyses of continental groups reflecting four ancestry clines, including individuals of African (AFR; maximum $n = 119,589$) and American (AMR; $n = 286,026$) recently admixed ancestries primarily from the United States; individuals of East Asian ancestries (EAS; $n = 296,438$) primarily from the United States, People's Republic of China and Japan; and individuals of European ancestries (EUR; $n = 2,669,029$) from the United States, Europe and Australia (see Extended Data Fig. 1 and Supplementary Note). Smoking phenotypes were selected to represent different stages of tobacco use and addiction, including initiation, the onset of regular use, amount smoked and cessation. Measures of onset included whether an individual ever smoked regularly (smoking initiation (SmkInit); $n = 3,383,199$) and the age at which the individual began smoking regularly (AgeSmk; $n = 728,826$). Amount smoked among current and former regular smokers was measured as cigarettes smoked per day (CigDay; $n = 784,353$). Smoking cessation (SmkCes; $n = 1,400,535$) contrasted current versus former smokers. Alcohol use was widely available across most studies, measured as drinks per week (DrnkWk; $n = 2,965,643$).

## Multi-ancestry meta-analysis

Using our multi-ancestry meta-analysis, we identified 2,143 associated loci across all phenotypes (sentinel variant $P < 5 \times 10^{-9}$), with 3,823 independently associated variants (Extended Data Fig. 2, Supplementary Tables 2 and 3 and Supplementary Figs. 2 and 3). Of these, 1,346 loci and 2,486 independent variants were associated with SmkInit, 33 loci (39 variants) with AgeSmk, 140 loci (243 variants) with CigDay, 128 loci (206 variants) with SmkCes and 496 loci (849 variants) with DrnkWk. Approximately 64% ($n = 1,364$) of loci were phenotype-specific, five loci were associated with all four smoking phenotypes but not with DrnkWk, and five loci were associated with all five phenotypes. All sentinel variants within identified loci had high posterior probabilities that their effect would replicate in a sufficiently powered study according to a trans-ancestry extension of our GWAS cross-validation technique[6]. Only 17 sentinel variants (0.7%) had such posterior probabilities of less than 0.99 and were therefore removed from the counts above and from further consideration (additional details on these 17 variants are shown in Supplementary Fig. 4).

Inclusion of diverse ancestry may improve the discovery of new variants through a combination of increased genetic variation, larger sample sizes and improved fine-mapping due to diverse patterns of linkage disequilibrium (LD). We quantified gains in power from the use of our multi-ancestry model over a simpler ancestry-naive fixed-effects model excluding the ancestry meta-regression. Comparing the number of associated variants, we found 721 additional independent variants that were identified only by the multi-ancestry meta-regression analysis. Both sets of models were fit to the same data, such that the larger number of significantly associated variants identified with the multi-ancestry model indicates increased power from accounting for axes of genetic variation and residual heterogeneity. Included among these 721 were newly associated variants in genes related to nervous system function (for example, *NRXN1*) including glutamatergic (*GRIN2A*) neurotransmission, which is of relevance to neurocircuitry in addiction[7,8].

To isolate likely causal variants, we used a fine-mapping procedure (see Supplementary Note) that leverages variation in LD across ancestry groups to construct 90% credible intervals. We identified 597 loci (27.9%) in which the 90% credible intervals included fewer than five

A list of authors and their affiliations appears online. ✉e-mail: dajiang.liu@psu.edu; vrieze@umn.edu

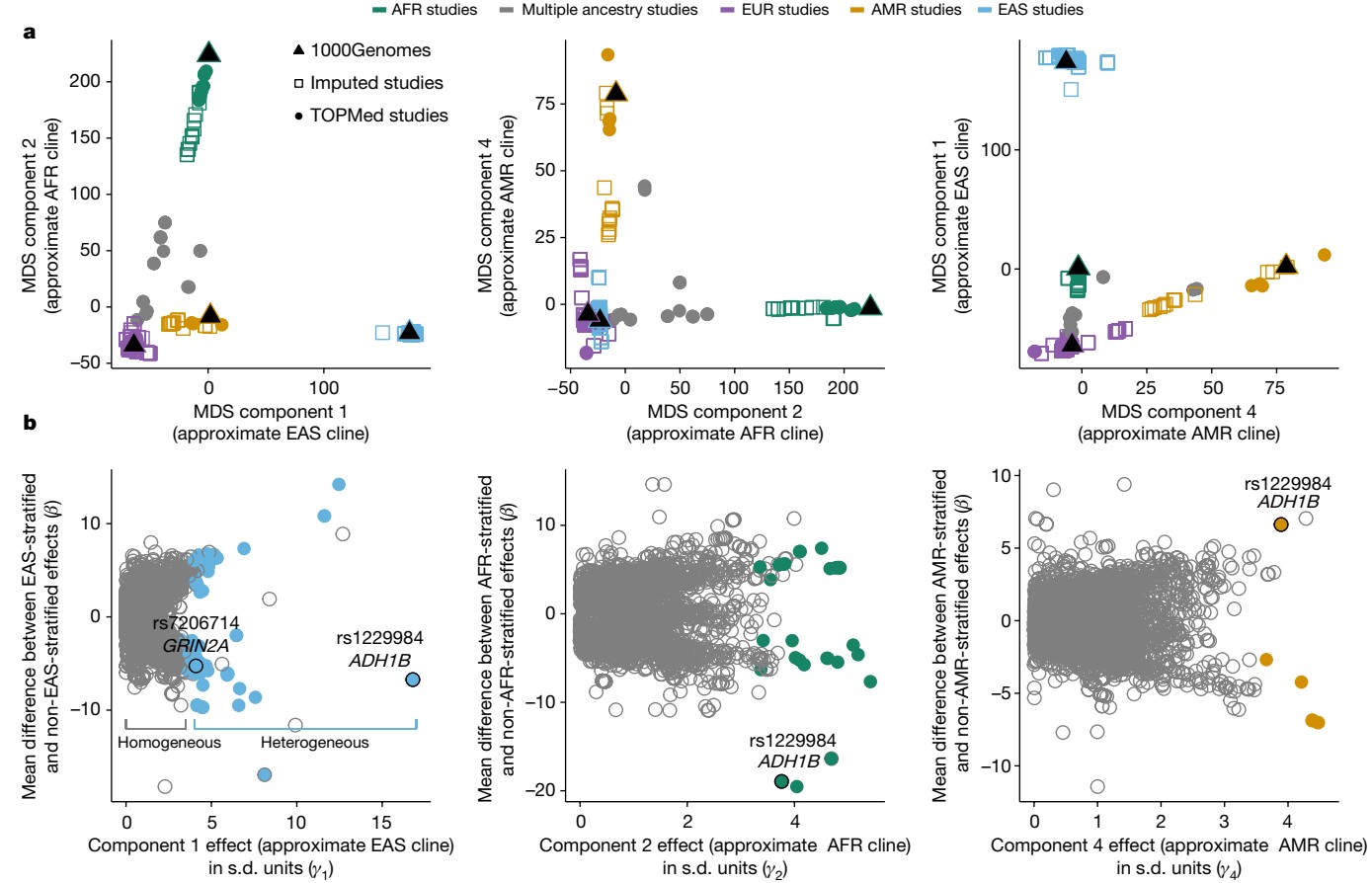

**Fig. 1 | Ancestry composition and effect size moderation. a**, Ancestry compositions of contributing studies (each point is a study). Colours are coded by primary ancestry of individuals in the cohort. Studies with less than 90% of individuals assignable to a single ancestry group are shown in grey. Ancestry component 3 was a north–south EUR cline, which was omitted here as we did not conduct meta-analyses stratified by northern versus southern Europe. TOPMed, Trans-Omics for Precision Medicine. **b**, Extent of effect size moderation as a function of the same ancestry dimensions as shown in **a**. The full moderation results are in Supplementary Table 2. Each point in **b** represents an independent variant with the standardized MDS component coefficient from our trans-ancestry models (that is, $\gamma$) along the $x$ axes, and the corresponding mean difference in effect sizes ($\beta$) for the ancestry-stratified meta-analysis of the given ancestry versus all other ancestries along the $y$ axes. The grey circles indicate variants showing little to no evidence of effect size heterogeneity across ancestry, whereas the coloured circles represent variants with adequate evidence of effect size heterogeneity. The plots highlight that the majority of variants have similar effect sizes across all ancestry clines, with some potentially interesting exceptions in which the variant effects sizes differ substantially between ancestry clines.

variants, including 192 loci (9.0%) with a single fine-mapped variant. Overall, credible intervals contained medians of 9–19 variants and median spans of 32–78 kb across phenotypes (Supplementary Table 4). Compared with the EUR-stratified GWAS (described in the next section), the trans-ancestry fine-mapping increased the number of 90% credible intervals containing fewer than five variants by 27.6%, and containing a single variant by 41.2%. Across all 2,143 loci, 1,330 (62.1%) loci had a reduced number of variants in the credible intervals in the multi-ancestry analysis. To determine the gain in resolution attributable to increased sample size (versus LD differences), we 'downsampled' the multi-ancestry analysis by removing EUR ancestry cohorts until the total sample size was approximately equal to that of the EUR-stratified analysis and regenerated fine-mapping results. Using the 1,330 loci with improved resolution in multi-ancestry analysis, we found that the credible intervals were reduced from a median of 22 variants in the EUR-stratified analysis to 12 variants in the downsampled multi-ancestry analysis, suggesting that approximately 55% of the observed improvement in fine-mapping is attributable to larger multi-ancestry sample sizes alone. These findings highlight the utility of both increased sample size and diverse ancestry in fine-mapping variants for these complex behavioural phenotypes. To characterize genes prioritized from fine-mapping, we conducted a series of functional enrichment analyses. We first selected intervals fine-mapped to fewer than five variants from the multi-ancestry results and mapped each variant to the nearest gene to identify 'high-priority' genes. Relative to genes mapped from variants with posterior inclusion probabilities (PIP) < 0.01, the high-priority genes were enriched across brain and nerve tissues (Extended Data Fig. 3a and Supplementary Table 5). Within the brain, cell-type enrichment of the high-priority genes was observed for projecting glutamatergic neurons from the cortex, hippocampus and amygdala (telencephalon excitatory projection neurons) and projection GABA neurons from medium spiny neurons of the striatum (telencephalon inhibitory projecting neurons), along with neurons in various subcortical structures such as the hypothalamus and midbrain, consistent with aspects of the mesolimbic theory of addiction[7,8] (Extended Data Fig. 3b). Finally, these high-priority genes that were strongly associated with substance use were enriched in gene pathways related to neurogenesis, neuronal development, neuronal differentiation and synaptic function. The neurodevelopmental aspect of the high-priority genes could indicate a role for these genes in processes that predispose individuals to risk of substance use and/or may contribute to brain circuit rewiring during drug use.

The multi-ancestry meta-analysis method also allowed for tests of whether a variant effect size differed (that is, was moderated) by

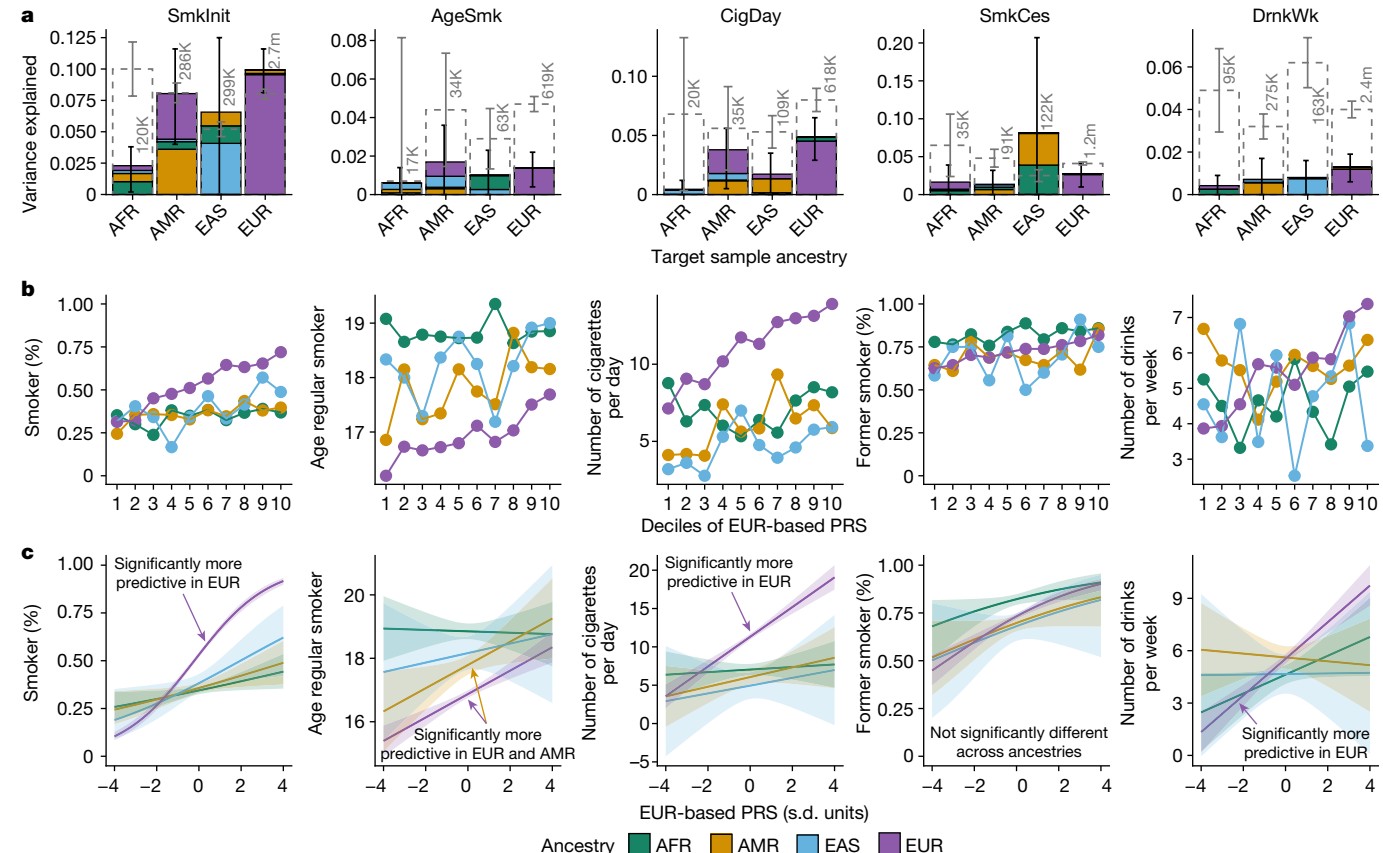

**Fig. 2 | Within-ancestry and across-ancestry performance of polygenic scores in an independent target sample (Add Health[35]). a**, Incremental variance explained for each target ancestry group. The colour of the stacked bars indicates the ancestry from which the polygenic score was derived; the total height of each set of the stacked bars (and 95% confidence intervals) correspond to the total variance explained by all four ancestry-stratified scores combined. For example, in the target EUR subsample, non-EUR polygenic scores add little over and above the EUR score. Note that some comparisons are underpowered to detect differences in predictive accuracy across ancestry (see Supplementary Note). Heritabilities, estimated by LD score regression, of each phenotype–ancestry combination are depicted by the grey dashed bar (with 95% confidence intervals) and corresponding sample sizes; these represent the maximum expected accuracy of the polygenic risk score (PRS). **b**, The manner in which the phenotype mean in the target sample changes as a function of the EUR PRS deciles. **c**, Results from an interaction model, in which each phenotype was modelled as a function of an interaction between the EUR-based PRS and target ancestry (coded as a factor with EUR ancestry as the reference and scores scaled within ancestry). The bands around each line denote the 95% confidence intervals. Significant interactions are noted with text. Using SmkInit as an example, the purple line represents the predicted proportion of regular smokers as a function of the EUR PRS in the EUR subsample of Add Health, the blue lines show the predicted proportion of regular smokers by standard deviation of the EUR PRS in the EAS subsample, and so on. In this case, the magnitude of the association between the EUR-based PRS and SmkInit (that is, the slope) was significantly greater in the EUR target ancestry than all other ancestries. Full PRS results are in Supplementary Table 12.

ancestry along four ancestry dimensions estimated from multidimensional scaling (MDS) of allele frequencies from each participating study (Fig. 1a). Roughly, the first axis represents an EAS ancestry cline, the second axis an AFR cline, the third a EUR cline (north to south EUR) and the fourth an AMR cline. There was minimal evidence of effect size moderation by ancestry for most independent variants, ranging from 76.6% (187 variants) in CigDay to 85.0% (175 variants) in SmkCes. Another 7.7–18.1% showed modest evidence for moderation. Finally, roughly 3.6% of all independent variants, reflecting 136 variants from 84 distinct loci, showed strong evidence of effect size moderated by ancestry (complete results are shown in Supplementary Table 2). Comparisons between the variants with strong evidence for effect size moderation by ancestry and those with no evidence suggested that the identification of these 136 variants was not driven to a large extent by differences in imputation quality, LD scores or Fst (fixation index) across ancestries.

Across phenotypes, 88 of these 136 variants showed moderation by the first axis of ancestry variation (approximate EAS cline; Fig. 1b, left), 29 variants by the second axis (approximate AFR cline; Fig. 1b, middle)

and 10 variants by the fourth axis (approximate AMR cline; Fig. 1b, right). Nine variants showed differences in effect size moderated by the third axis (EUR cline). Only the effect of one variant was moderated by three or more ancestry clines (EAS, AFR and AMR): rs1229984, a missense variant in the alcohol dehydrogenase gene *ADH1B*, which has been shown to be protective against alcohol consumption[9]. An increase on any of these clines was associated with a reduced effect size of this allele, on average. For example, if there are two people who both carry one copy of the protective T allele for this variant but are separated by 1 s.d. on MDS component 1 (EAS cline), the person with a lower value on that MDS cline would be expected to drink 0.3 fewer drinks than the person with a higher MDS value, despite the same rs1229984 genotype in *ADH1B*.

To further evaluate causal genes and relevant tissues through which associated variants may be operating, we applied a trans-ancestry transcriptome-wide association study (TWAS) analysis to each phenotype across 49 tissues derived from the GTEx Consortium[10]. Using a *P* value threshold Bonferroni-corrected for the total number of genes and tissues within a phenotype, we found 1,167 genes significantly

associated with SmkInit, 21 genes with AgeSmk, 203 genes with CigDay, 188 genes with SmkCes and 504 genes with DrnkWk (resulting in 1,705 unique genes across phenotypes; Supplementary Table 6). For each of our five phenotypes, matrix decomposition parallel analysis[11] of the per-tissue $P$ value correlation matrix suggested two components: one explaining 53.7–55.2% of the variance in $P$ values, and another explaining 3.5–3.8% of the variance in $P$ values. Similar loading patterns were observed for all phenotypes such that all tissues loaded strongly (all loadings > 0.12) on the first component, suggesting that it represents a general effect across tissues, whereas only brain tissues had strong loadings on the second component (all loadings > 0.12), indicating the importance of brain-specific gene expression effects for these tobacco and alcohol use phenotypes. Pathway enrichment analyses of the TWAS-associated genes identified 1,029 unique gene pathways across phenotypes that were broadly enriched across tissues (Supplementary Table 7), including many of obvious relevance to neurotransmission and neurodevelopment.

To further illustrate several variants within genes of interest, we integrated findings described above to select variants for which there was evidence of association across analytic methods and for which the availability of diverse ancestries was clearly relevant. Illustrative variants were chosen in a similar way as described for the enrichment analyses above: (1) we extracted variants from multi-ancestry fine-mapped credible intervals containing less than five variants, and (2) we cross-referenced the resulting variants with the multi-ancestry TWAS *cis*-expression quantitative trait loci and their significantly associated genes. We highlight five of the 52 genes that resulted from this process.

We found the nicotinic gene cluster *CHRNA5–A3–B4* to be significantly associated with SmkInit[12] with a fine-mapped 90% credible interval that shrank from 53 variants in EUR-stratified results to just two variants in multi-ancestry results (rs2869055 and rs28438420; Supplementary Table 4). These variants are not in high LD ($r^2 = 0.31$ for both variants) with the well-known variant rs16969968 in this gene cluster. By contrast, this locus was fine-mapped to two variants in high LD with rs16969968 for CigDay ($r^2 = 0.84$ and 0.86), suggesting that the variants underlying this signal for smoking initiation may be distinct from those for cigarettes per day. We also found a novel association between SmkInit and *CACNA1B*, which encodes a voltage-gated calcium channel ($Ca_v2.2$) that controls neuronal neurotransmitter release and has been associated with cocaine reinstatement[13], increased aggression and vigilance, and reduced startle and exploration[14]. *CACNA1B* is linked to multiple psychiatric disorders, including schizophrenia, bipolar disorder and autism spectrum disorders[15–17].

CigDay was associated with variants in neurturin (*NRTN*), a type of glial cell line-derived neurotrophic factor involved in the development and survival of dopamine neurons[18]. This gene has been studied in relation to Parkinson disease for its potential to restore dopamine neurocircuitry[19]. Likewise, *PAK6* was another novel gene strongly associated with CigDay in TWAS results and was fine-mapped to just three variants in the 90% credible interval. *PAK6* encodes a p21-activated kinase that is highly expressed in the striatum and hippocampus, has been implicated in the migration of GABAergic interneurons[20] as well as the modulation of dopaminergic neurotransmission[21], and is involved in locomotor activity and cognitive function[22]. *PAK6* has been robustly associated with schizophrenia[23] and neurodegenerative diseases[24,25], such as Parkinson disease and Alzheimer disease, further highlighting its role in synaptic changes. Finally, we found a novel association between *ECE2* and DrnkWk. *ECE2* is involved in cortical development[26] as well as the processing of several neuroendocrine peptides, including neurotensin and substance P[27], and may also have a role in amyloid-β processing[28]. *ECE2* also generates peptides such as BAM 12 (which shows $\kappa$-opioid receptor selectivity) and BAM 22 (which shows $\mu$-opioid receptor selectivity), suggesting a link with pain transmission[27].

## Genetic correlation and polygenic scores

To evaluate heritability, genetic correlation and polygenic scoring, we generated ancestry-stratified GWAS meta-analysis results for each of the four continental groups: AFR, AMR, EAS and EUR (Supplementary Table 2 lists ancestry-stratified loci). Heritability and cross-phenotype genetic correlations were generally similar in sign and modest in magnitude in each ancestry (Fig. 2a and Supplementary Tables 8 and 9). Smoking phenotypes were moderately genetically correlated with each other ($|r_g| = 0.30–0.63$) and with DrnkWk ($|r_g| = 0.16–0.27$). Genetic correlations for the same phenotype between each of the largest contributing cohorts and all remaining cohorts (restricted to EUR ancestries only) were generally high for each smoking phenotype (mean $r_g$ of 0.93) and DrnkWk (mean $r_g$ of 0.72), indicating that these measures were reliable across cohorts (Supplementary Table 9).

To characterize the multifactorial genetic aetiology of tobacco and alcohol use, we computed genetic correlations of our EUR-stratified results with 1,141 medical, biomarker and behavioural phenotypes from the UK Biobank[29] (Supplementary Tables 10 and 11). An affinity propagation clustering algorithm[30] was used to aid interpretability by grouping UK Biobank phenotypes such that each of the five current phenotypes were exemplars (Supplementary Fig. 5). SmkInit and AgeSmk clustered together, as did SmkCes and CigDay, with all four forming a broad higher-level smoking cluster. Phenotypes with high positive genetic correlations with SmkInit included addiction to any substance, neighbourhood material deprivation, diagnosis of chronic obstructive pulmonary disease, and a negative correlation with age at first sexual intercourse ($|r_g| = 0.57–0.64$). For AgeSmk, the largest genetic correlations were with reproductive phenotypes such as age at first birth ($r_g = 0.69–0.71$) and measures of years of education and attainment ($r_g = 0.58–0.69$). CigDay and SmkCes were most highly positively correlated with respiratory and cardiovascular diseases and cancers ($r_g = 0.52–0.72$), highlighting their genetic link to adverse disease outcomes. Finally, DrnkWk was most strongly correlated with problematic drinking behaviours ($r_g = 0.52–0.70$), indicating extensive overlap in the genetic architecture of DrnkWk and measures of alcohol use, problems and alcohol use disorder. This is consistent with previous findings of strong but imperfect genetic correlations (for example, $r_g = 0.8$) between alcohol consumption and alcohol use disorder from large-scale GWAS[31,32]. We note, however, that genetic correlations can be difficult to interpret[33,34] as they may be affected by genetic confounding, mediation effects or sampling bias.

We used the ancestry-stratified meta-analysis results to construct ancestry-specific polygenic risk scores in Add Health[35], an independent target sample of individuals of diverse ancestries from the United States ($n = 2,199$ AFR, 1,132 AMR, 525 EAS and 6,092 EUR). To evaluate within-ancestry and across-ancestry performance of polygenic scores, we iteratively fit a multiple regression model and evaluated the incremental predictive accuracy of each ancestry-based score, over and above scores already entered into the model (that is, first including the AMR-based score, then adding the AFR-based, EAS-based and EUR-based scores one at a time to evaluate incremental prediction accuracy). EUR-based scores in EUR ancestries outperformed ancestry-matched scores in non-EUR ancestries (Fig. 2a) and showed significantly stronger associations with most phenotypes in EUR ancestries than in non-EUR ancestries (described by decile plots and tested by modelling an interaction between the EUR-based polygenic risk score and the target sample ancestry group), consistent with expectations[36] (Fig. 2b,c). For each ancestry and phenotype, the EUR-based score on its own outperformed the ancestry-matched score on its own (Supplementary Table 12). These results highlight the relative utility of current polygenic scores for EUR ancestries versus all others. In interpreting these results, however, we note that some comparisons may be underpowered to identify differences in the variance explained by polygenic scores between ancestries. Finally, EUR-based scores

overpredicted tobacco and alcohol use for individuals of non-EUR ancestry and underpredicted for individuals of EUR ancestry, although this prediction bias is readily eliminated through statistical correction with genetic principal components.

## Summary

Tobacco and alcohol use are heritable behaviours that can be radically affected by environmental factors, including cultural context[37] and public health policies[38,39]. Despite this, we found that a large majority of associated genetic variants showed homogeneous effect size estimates across diverse ancestries, suggesting that the genetic variants associated with substance use affect such individuals similarly. The limited extent of variant effect size heterogeneity, coupled with similar heritability estimates and cross-trait genetic correlations, indicates that the genetic architecture underlying substance use is not markedly different across ancestries. There are some potentially interesting exceptions of ancestrally heterogeneous effects in genes such as *ADH1B* and *CACNA1B*. By contrast, polygenic scores generally performed well in EUR ancestries but with mixed-to-limited results in other ancestries, suggesting that portability of such scores across ancestries remains challenging, even when discovery sample sizes across all ancestries are more than 100,000. Explanations for this apparent discrepancy have been proposed[40], but more stringent and sensitive tests will be required to draw strong conclusions about such patterns of heredity.

Most individuals of EUR, AFR and AMR ancestries in the current study live in the United States and Europe and share somewhat similar environments regarding tobacco and alcohol availability and policies surrounding use of these substances, and all included individuals were adults. Further increases in genetic diversity and consideration of environmental moderators, including cultural factors, will continue to add to our understanding of the genetic architecture of both substance use and related behaviours and diseases.

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

Gretchen R. B. Saunders[1,170], Xingyan Wang[2,170], Fang Chen[2,170], Seon-Kyeong Jang[1,170], Mengzhen Liu[1,170], Chen Wang[2,170], Shuang Gao[2], Yu Jiang[3], Chachrit Khunsriraksakul[2], Jacqueline M. Otto[1], Clifton Addison[4], Masato Akiyama[5,6], Christine M. Albert[7,8], Fazil Aliev[9], Alvaro Alonso[10], Donna K. Arnett[11], Allison E. Ashley-Koch[12,13], Aneel A. Ashrani[14], Kathleen C. Barnes[15,16], R. Graham Barr[17], Traci M. Bartz[18,19], Diane M. Becker[20], Lawrence F. Bielak[21], Emelia J. Benjamin[22,23], Joshua C. Bis[18], Gyda Bjornsdottir[24], John Blangero[25], Eugene R. Bleecker[26], Jason D. Boardman[27], Eric Boerwinkle[28], Dorret I. Boomsma[29], Meher Preethi Boorgula[15], Donald W. Bowden[30], Jennifer A. Brody[18], Brian E. Cade[31,32,33], Daniel I. Chasman[8], Sameer Chavan[15], Yii-Der Ida Chen[34], Zhengming Chen[35,36], Iona Cheng[37,38], Michael H. Cho[39,40], Hélène Choquet[41], John W. Cole[42,43], Marilyn C. Cornelis[44], Francesco Cucca[45], Joanne E. Curran[25], Mariza de Andrade[46], Danielle M. Dick[9], Anna R. Docherty[47,48,49], Ravindranath Duggirala[25], Charles B. Eaton[50], Marissa A. Ehringer[51,52], Tõnu Esko[53], Jessica D. Faul[54], Lilian Fernandes Silva[55], Edoardo Fiorillo[56], Myriam Fornage[28,57], Barry I. Freedman[58], Maiken E. Gabrielsen[59], Melanie E. Garrett[12,13], Sina A. Gharib[18,60,61], Christian Gieger[62], Nathan Gillespie[48], David C. Glahn[63], Scott D. Gordon[64], Charles C. Gu[65], Dongfeng Gu[66], Daniel F. Gudbjartsson[24,67], Xiuqing Guo[68], Jeffrey Haessler[69], Michael E. Hall[70], Toomas Haller[53], Kathleen Mullan Harris[71], Jiang He[72,73], Pamela Herd[74], John K. Hewitt[51,75], Ian Hickie[76], Bertha Hidalgo[77], John E. Hokanson[78], Christian Hopfer[79], JoukeJan Hottenga[29], Lifang Hou[44], Hongyan Huang[80,81], Yi-Jen Hung[82], David J. Hunter[83], Kristian Hveem[59,84,85], Shih-Jen Hwang[86], Chii-Min Hwu[87], William Iacono[1], Marguerite R. Irvin[77], Yun Ho Jee[80], Eric O. Johnson[88,89], Yoonjung Y. Joo[44,90], Eric Jorgenson[91], Anne E. Justice[92,93], Yoichiro Kamatani[5,94], Robert C. Kaplan[69,95], Jaakko Kaprio[96], Sharon L. R. Kardia[21], Matthew C. Keller[51,75], Tanika N. Kelly[72,73], Charles Kooperberg[19,69], Tellervo Korhonen[96], Peter Kraft[80,81], Kenneth Krauter[97], Johanna Kuusisto[98,99], Markku Laakso[98], Jessica Lasky-Su[100], Wen-Jane Lee[101], James J. Lee[1], Daniel Levy[86], Liming Li[102], Kevin Li[103], Yuqing Li[37], Kuang Lin[35], Penelope A. Lind[104,105,106], Chunyu Liu[107], Donald M. Lloyd-Jones[108], Sharon M. Lutz[109,110], Jiantao Ma[86,111], Reedik Mägi[52], Ani Manichaikul[112], Nicholas G. Martin[64], Ravi Mathur[88], Nana Matoba[5,113], Patrick F. McArdle[114], Matt McGue[1], Matthew B. McQueen[115], Sarah E. Medland[104], Andres Metspalu[53], Deborah A. Meyers[26], Iona Y. Millwood[35,36], Braxton D. Mitchell[114,116], Karen L. Mohlke[117], Matthew Moll[39,40], May E. Montasser[114], Alanna C. Morrison[28], Antonella Mulas[56], Jonas B. Nielsen[59,118], Kari E. North[93], Elizabeth C. Oelsner[17], Yukinori Okada[119,120,121,122], Valeria Orrù[56], Nicholette D. Palmer[30], Teemu Palviainen[96], Anita Pandit[103], S. Lani Park[123], Ulrike Peters[69,124], Annette Peters[125,126,127], Patricia A. Peyser[21], Tinca J. C. Polderman[128,129], Nicholas Rafaels[15], Susan Redline[31,32,130], Robert M. Reed[131], Alex P. Reiner[69,124], John P. Rice[132], Stephen S. Rich[112], Nicole E. Richmond[78], Carol Roan[133], Jerome I. Rotter[68], Michael N. Rueschman[31], Valgerdur Runarsdottir[134], Nancy L. Saccone[135], David A. Schwartz[136], Aladdin H. Shadyab[137], Jingchunzi Shi[138], Suyash Shringarpure[138], Kamil Sicinski[133], Anne Heidi Skogholt[59], Jennifer A. Smith[21,54], Nicholas L. Smith[124,139,140], Nona Sotoodehnia[18,141], Michael C. Stallings[51,75], Hreinn Stefansson[24], Kari Stefansson[24,142], Jerry A. Stitzel[51], Xiao Sun[72], Moin Syed[1], Ruth Taylor-Singer[143], Amy E. Taylor[144,145,146], Kent D. Taylor[68], Marilyn J. Telen[12], Khanh K. Thai[41], Hemant Tiwari[147], Constance Turman[80,81], Thorarinn Tyrfingsson[134], Tamara L. Wall[148], Robin G. Walters[35,36], David R. Weir[54], Scott T. Weiss[100], Wendy B. White[149], John B. Whitfield[64], Kerri L. Wiggins[150], Gonneke Willemsen[29], Cristen J. Willer[151,152,153], Bendik S. Winsvold[59,154,155], Huichun Xu[114], Lisa R. Yanek[20], Jie Yin[41], Kristin L. Young[156], Kendra A. Young[78], Bing Yu[28], Wei Zhao[21], Wei Zhou[153,157], Sebastian Zöllner[158,159], Luisa Zuccolo[144,146,160], 23andMe Research Team*, The Biobank Japan Project*, Chiara Batini[161], Andrew W. Bergen[162,163], Laura J. Bierut[132], Sean P. David[164,165], Sarah A. Gagliano Taliun[166,167,168], Dana B. Hancock[88], Bibo Jiang[2], Marcus R. Munafò[144,145,169], Thorgeir E. Thorgeirsson[24], Dajiang J. Liu[2,171 ✉] & Scott Vrieze[1,171 ✉]

[1]Department of Psychology, University of Minnesota, Minneapolis, MN, USA. [2]Department of Public Health Sciences, Penn State College of Medicine, Hershey, PA, USA. [3]Department of Epidemiology & Population Health at Stanford University, Stanford, CA, USA. [4]Jackson Heart Study (JHS) Graduate Training and Education Center (GTEC), Department of Epidemiology and Biostatistics, School of Public Health, Jackson State University, Jackson, MS, USA. [5]Laboratory for Statistical and Translational Genetics, RIKEN Center for Integrative Medical Sciences, Yokohama, Japan. [6]Department of Ocular Pathology and Imaging Science, Kyushu University Graduate School of Medical Sciences, Fukuoka, Japan. [7]Smidt Heart Institute, Cedars-Sinai Medical Center, Los Angeles, CA, USA. [8]Division of Preventive Medicine, Brigham and Women's Hospital and Harvard Medical School, Boston, MA, USA. [9]Department of Psychiatry, Rutgers Robert Wood Johnson Medical School, New Brunswick, NJ, USA. [10]Department of Epidemiology, Rollins School of Public Health, Emory University, Atlanta, GA, USA. [11]Dean's Office and Department of Epidemiology, College of Public Health, University of Kentucky, Lexington, KY, USA. [12]Department of Medicine and Duke Comprehensive Sickle Cell Center, Duke University School of Medicine, Durham, NC, USA. [13]Duke Molecular Physiology Institute, Duke University School of Medicine, Durham, NC, USA. [14]Division of Hematology, Department of Medicine, Mayo Clinic College of Medicine and Science, Rochester, MN, USA. [15]Division of Biomedical Informatics & Personalized Medicine, University of Colorado Anschutz Medical Campus, Aurora, CO, USA. [16]Tempus, Chicago, IL, USA. [17]Department of Medicine, Columbia University Medical Center, New York, NY, USA. [18]Cardiovascular Health Research Unit, Department of Medicine, University of Washington, Seattle, WA, USA. [19]Department of Biostatistics, University of Washington, Seattle, WA, USA. [20]Department of Medicine, Johns Hopkins University School of Medicine, Baltimore, MD, USA. [21]Department of Epidemiology, School of Public Health, University of Michigan, Ann Arbor, MI, USA. [22]Department of Medicine, Boston Medical Center, Boston University School of Medicine, Boston, MA, USA. [23]Department of Epidemiology, Boston University School of Public Health, Boston, MA, USA. [24]deCODE Genetics/Amgen, Inc., Reykjavik, Iceland. [25]Department of Human Genetics and South Texas Diabetes and Obesity Institute, University of Texas Rio Grande Valley School of Medicine, Brownsville, TX, USA. [26]Department of Medicine, University of Arizona, Tucson, AZ, USA. [27]Institute of Behavioral Science, University of Colorado Boulder, Boulder, CO, USA. [28]Human Genetics Center, Department of Epidemiology, Human Genetics, and Environmental Sciences, School of Public Health, The University of Texas Health Science Center at Houston, Houston, TX, USA. [29]Netherlands Twin Register, Dept Biological Psychology, Vrije Universiteit Amsterdam, Amsterdam, The Netherlands. [30]Department of Biochemistry, Wake Forest School of Medicine, Winston-Salem, NC, USA. [31]Division of Sleep and Circadian Disorders, Brigham and Women's Hospital, Boston, MA, USA. [32]Division of Sleep Medicine, Harvard Medical School, Boston, MA, USA. [33]Program in Medical and Population Genetics, Broad Institute, Cambridge, MA, USA. [34]Institute for Translational Genomics and Population Sciences, Department of Pediatrics, Lundquist Institute for Biomedical Innovation at Harbor-UCLA Medical Center, Torrance, CA, USA. [35]Clinical Trial Service Unit and Epidemiological Studies Unit, Nuffield Department of Population Health, University of Oxford, Oxford, UK. [36]MRC Population Health Research Unit, Nuffield Department of Population Health, University of Oxford, Oxford, UK. [37]Department of Epidemiology & Biostatistics, University of California, San Francisco, CA, USA. [38]UCSF Helen Diller Family Comprehensive Cancer Center, University of California, San Francisco, CA, USA. [39]Channing Division of Network Medicine, Department of Medicine, Brigham and Women's Hospital, Boston, MA, USA. [40]Division of Pulmonary and Critical Care Medicine, Department of Medicine, Brigham and Women's Hospital, Boston, MA, USA. [41]Kaiser Permanente Northern California (KPNC), Division of Research, Oakland, CA, USA. [42]Department of Neurology, Baltimore Veterans Affairs Medical Center, Baltimore, MD, USA. [43]Division of Vascular Neurology, Department of Neurology, University of Maryland School of Medicine, Baltimore, MD, USA. [44]Department of Preventive Medicine, Northwestern University Feinberg School of Medicine, Chicago, IL, USA. [45]University of Sassari, Sassari SS, Italy. [46]Division of Clinical Trials and Biostatistics, Department of Quantitative Health Sciences, Mayo Clinic College of Medicine and Science, Rochester, MN, USA. [47]Department of Psychiatry, University of Utah School of Medicine, Salt Lake City, UT, USA. [48]Virginia Institute for Psychiatric and Behavioral Genetics, Virginia Commonwealth University, Virginia, USA. [49]Huntsman Mental Health Institute, Salt Lake City, UT, USA. [50]Department of Family Medicine, Brown University, Providence, RI, USA. [51]Institute for Behavioral Genetics, University of Colorado Boulder, Boulder, CO, USA. [52]Department of Integrative Physiology, University of Colorado Boulder, Boulder, CO, USA. [53]Institute of Genomics, University of Tartu, Tartu, Estonia. [54]Survey Research Center, Institute for Social Research, University of Michigan, Ann Arbor, MI, USA. [55]Institute of Clinical Medicine, Internal Medicine, University of Eastern Finland, Kuopio, Finland. [56]Istituto di Ricerca Genetica e Biomedica, Consiglio Nazionale delle Ricerche (CNR), Monserrato, Italy. [57]Brown Foundation Institute of Molecular Medicine, McGovern Medical School, University of Texas Health Science Center at Houston, Houston, TX, USA. [58]Department of Internal Medicine-Section on Nephrology, Wake Forest School of Medicine, Winston-Salem, NC, USA. [59]K.G. Jebsen Center for Genetic Epidemiology, Department of Public Health and Nursing, NTNU, Norwegian University of Science and Technology, Trondheim, Norway. [60]Division of Pulmonary, Critical Care, and Sleep Medicine, Department of Medicine, University of Washington, Seattle, WA, USA. [61]Center for Lung Biology, Department of Medicine, University of Washington, Seattle, WA, USA. [62]Research Unit Molecular Epidemiology, Institute of Epidemiology, Helmholtz Zentrum München, German Research Center for Environmental Health, Neuherberg, Germany. [63]Department of Psychiatry & Behavioral Sciences, Boston Children's Hospital & Harvard Medical School, Boston, MA, USA. [64]Genetic Epidemiology, QIMR Berghofer Medical Research Institute, Brisbane, Australia. [65]Division of Biostatistics, Washington University School of Medicine, St. Louis, MO, USA. [66]Department of Epidemiology and Key Laboratory of Cardiovascular Epidemiology, Fuwai Hospital, National Center for Cardiovascular Diseases, Chinese Academy of Medical Sciences and Peking Union Medical College, Beijing, China. [67]School of Engineering and Natural Sciences, University of Iceland, Reykjavik, Iceland. [68]The Institute for Translational Genomics and Population Sciences, Department of Pediatrics, The Lundquist Institute for Biomedical Innovation at Harbor-UCLA Medical Center, Torrance, CA, USA. [69]Public Health Sciences Division, Fred Hutchinson Cancer Research Center, Seattle, WA, USA. [70]Department of Medicine, University of Mississippi Medical Center, Jackson, MS, USA. [71]Department of Sociology and the Carolina Population Center, University of North Carolina, Chapel Hill, NC, USA. [72]Department of Epidemiology, Tulane University, New Orleans, LA, USA. [73]Translational Sciences Institute, Tulane University, New Orleans, LA, USA. [74]McCourt School of Public Policy, Georgetown University, Washington, DC, USA. [75]Department Of Psychology and Neuroscience, University of Colorado Boulder, Boulder, CO, USA. [76]Youth Mental Health & Technology Team, Brain and Mind Centre, University of Sydney, Sydney, Australia. [77]Department of Epidemiology, School of Public Health, University of Alabama at Birmingham, Birmingham, AL, USA. [78]Department of Epidemiology, University of Colorado Anschutz Medical Campus, Aurora, CO, USA. [79]Department of Psychiatry, University of Colorado Anschutz Medical Center, Denver, CO, USA. [80]Department of Epidemiology, Harvard T.H. Chan School of Public Health, Boston, MA, USA. [81]Program in Genetic Epidemiology and Statistical Genetics, Harvard T.H. Chan School of Public Health, Boston, MA, USA. [82]Institute of Preventive Medicine, National Defense Medical Center, New Taipei City, Taiwan. [83]Nuffield Department of Population Health, University of Oxford, Oxford, UK. [84]HUNT Research Center, Department of Public Health and Nursing, Faculty of Medicine and Health Sciences, Norwegian University of Science and Technology (NTNU), Trondheim, Norway. [85]Department of Research, Innovation and Education, St. Olavs Hospital, Trondheim University Hospital, Trondheim, Norway. [86]Population Sciences Branch, National Heart, Lung, and Blood Institute, National Institutes of Health, Bethesda, MD, USA. [87]Section of Endocrinology and Metabolism, Department of Medicine, Taipei Veterans General Hospital, Taipei, Taiwan. [88]GenOmics, Bioinformatics, and Translational Research Center, RTI International, Research Triangle Park, NC, USA. [89]Fellow Program, RTI International, Research Triangle Park, NC, USA.

[90]Institute of Data Science, Korea University, Seoul, South Korea. [91]Regeneron Genetics Center, Tarrytown, NY, USA. [92]Department of Population Health Sciences, Geisinger, Danville, PA, USA. [93]Department of Epidemiology, University of North Carolina at Chapel Hill, Chapel Hill, NC, USA. [94]Laboratory of Complex Trait Genomics, Department of Computational Biology and Medical Sciences, Graduate School of Frontier Sciences, The University of Tokyo, Tokyo, Japan. [95]Department of Epidemiology and Population Health, Albert Einstein College of Medicine, Bronx, NY, USA. [96]Institute for Molecular Medicine Finland - FIMM, University of Helsinki, Helsinki, Finland. [97]Department of Molecular, Cellular and Developmental Biology, University of Colorado, Boulder, CO, USA. [98]Institute of Clinical Medicine, Internal Medicine, University of Eastern Finland and Kuopio University Hospital, Kuopio, Finland. [99]Center for Medicine and Clinical Research, Kuopio University Hospital, Kuopio, Finland. [100]Brigham and Women's Hospital, Department of Medicine, Channing Division of Network Medicine, Boston, MA, USA. [101]Department of Medical Research, Taichung Veterans General Hospital, Taichung City, Taiwan. [102]Department of Epidemiology and Biostatistics, School of Public Health, Peking University Health Science Center, Beijing, China. [103]Center for Statistical Genetics, Department of Biostatistics, University of Michigan, Ann Arbor, MI, USA. [104]Psychiatric Genetics, QIMR Berghofer Medical Research Institute, Brisbane, Australia. [105]School of Biomedical Sciences, Faculty of Medicine, University of Queensland, Brisbane, Australia. [106]School of Biomedical Sciences, Queensland University of Technology, Brisbane, Australia. [107]Department of Biostatistics, Boston University School of Public Health, Boston, MA, USA. [108]Departments of Preventive Medicine, Medicine, and Pediatrics, Northwestern University Feinberg School of Medicine, Chicago, IL, USA. [109]Department of Population Medicine, Harvard Pilgrim Health Care Institute, Boston, MA, USA. [110]Department of Biostatics, Harvard T.H. Chan School of Public Health, Boston, MA, USA. [111]Division of Nutrition Epidemiology and Data Science, Friedman School of Nutrition Science and Policy, Tufts University, Boston, MA, USA. [112]Center for Public Health Genomics, Department of Public Health Sciences, University of Virginia School of Medicine, Charlottesville, VA, USA. [113]Department of Genetics, UNC Neuroscience Center, University of North Carolina at Chapel Hill, Chapel Hill, NC, USA. [114]Division of Endocrinology, Diabetes and Nutrition, Department of Medicine, University of Maryland School of Medicine, Baltimore, MD, USA. [115]Department of Integrative Physiology, University of Colorado, Boulder, CO, USA. [116]Geriatrics Research and Education Clinical Center, Baltimore Veterans Administration Medical Center, Baltimore, MD, USA. [117]Department of Genetics, University of North Carolina, Chapel Hill, NC, USA. [118]Department of Internal Medicine, Division of Cardiovascular Medicine, University of Michigan, Ann Arbor, MI, USA. [119]Laboratory for Systems Genetics, RIKEN Center for Integrative Medical Sciences, Yokohama, Japan. [120]Department of Statistical Genetics, Osaka University Graduate School of Medicine, Suita, Japan. [121]Laboratory of Statistical Immunology, Immunology Frontier Research Center (WPI-IFReC), Osaka University, Suita, Japan. [122]Department of Genome Informatics, Graduate School of Medicine, the University of Tokyo, Tokyo, Japan. [123]Population Sciences of the Pacific Program, University of Hawaii Cancer Center, Honolulu, HI, USA. [124]Department of Epidemiology, University of Washington, Seattle, WA, USA. [125]Institute of Epidemiology, Helmholtz Zentrum München, German Research Center for Environmental Health, Neuherberg, Germany. [126]Institute for Medical Information Processing, Biometry and Epidemiology, Ludwig Maximilians University Munich, Munich, Germany. [127]German Centre for Cardiovascular Research, DZHK, Partner Site Munich, Munich, Germany. [128]Department of Clinical Developmental Psychology, Vrije Universiteit, Amsterdam, The Netherlands. [129]Department of Child and Adolescent Psychiatry, Amsterdam UMC, Amsterdam, The Netherlands. [130]Division of Pulmonary, Critical Care, and Sleep Medicine, Beth Israel Deaconess Medical Center, Boston, MA, USA. [131]Division of Pulmonary and Critical Care Medicine, Department of Medicine, University of Maryland School of Medicine, Baltimore, MD, USA. [132]Department of Psychiatry, Washington University School of Medicine, St. Louis, MO, USA. [133]Center for Demography of Health and Aging, University of Wisconsin-Madison, Madison, WI, USA. [134]SAA-National Center of Addiction Medicine, Vogur Hospital, Reykjavik, Iceland. [135]Department of Genetics, Washington University School of Medicine, St. Louis, MO, USA. [136]Division of Pulmonary Sciences and Critical Care Medicine; Department of Medicine and Immunology, University of Colorado, Aurora, CO, USA. [137]Herbert Wertheim School of Public Health and Human Longevity Science, University of California, San Diego, La Jolla, CA, USA. [138]23andMe, Inc, Sunnyvale, CA, USA. [139]Kaiser Permanente Washington Health Research Institute, Kaiser Permanente Washington, Seattle, WA, USA. [140]Seattle Epidemiologic Research and Information Center, Department of Veterans Affairs Office of Research and Development, Seattle, WA, USA. [141]Division of Cardiology, Department of Medicine, University of Washington, Seattle, WA, USA. [142]Faculty of Medicine, University of Iceland, Reykjavik, Iceland. [143]COPD Foundation, Washington, DC, USA. [144]MRC Integrative Epidemiology Unit, Population Health Sciences, University of Bristol, Bristol, UK. [145]National Institute for Health Research Biomedical Research Centre at the University Hospitals Bristol NHS Foundation Trust and the University of Bristol, Bristol, UK. [146]Department of Population Health Sciences, Bristol Medical School, University of Bristol, Bristol, UK. [147]Department of Biostatistics, School of Public Health, University of Alabama at Birmingham, Birmingham, AL, USA. [148]Department of Psychiatry, University of California San Diego, San Diego, CA, USA. [149]Jackson Heart Study Undergraduate Training and Education Center, Tougaloo College, Tougaloo, MS, USA. [150]Department of Medicine, University of Washington, Seattle, WA, USA. [151]Department of Internal Medicine, Division of Cardiology, University of Michigan, Ann Arbor, MI, USA. [152]Department of Human Genetics, University of Michigan, Ann Arbor, MI, USA. [153]Department of Computational Medicine and Bioinformatics, University of Michigan, Ann Arbor, MI, USA. [154]Department of Research and Innovation, Division of Clinical Neuroscience, Oslo University Hospital, Oslo, Norway. [155]Department of Neurology, Oslo University Hospital, Oslo, Norway. [156]Department of Epidemiology, Gillings School of Global Public Health, University of North Carolina at Chapel Hill, Chapel Hill, NC, USA. [157]Analytic and Translational Genetics Unit, Massachusetts General Hospital, Boston, MA, USA. [158]Department of Biostatistics, University of Michigan, Ann Arbor, MI, USA. [159]Department of Psychiatry, University of Michigan, Ann Arbor, MI, USA. [160]Health Data Science Centre, Fondazione Human Technopole, Milan, Italy. [161]Department of Population Health Sciences, University of Leicester, Leicester, UK. [162]Oregon Research Institute, Springfield, OR, USA. [163]BioRealm, LLC, Walnut, CA, USA. [164]Outcomes Research Network & Department of Family Medicine, NorthShore University HealthSystem, Evanston, IL, USA. [165]Department of Family Medicine, University of Chicago, Chicago, IL, USA. [166]Department of Medicine, Université de Montréal, Montréal, Québec, Canada. [167]Department of Neurosciences, Université de Montréal, Montréal, Québec, Canada. [168]Research Centre, Montréal Heart Institute, Montréal, Québec, Canada. [169]School of Psychological Science, University of Bristol, Bristol, UK. [170]These authors contributed equally: Gretchen R.B. Saunders, Xingyan Wang, Fang Chen, Seon-Kyeong Jang, Mengzhen Liu, Chen Wang. [171]These authors jointly supervised this work: Dajiang J. Liu, Scott Vrieze. *A list of authors and their affiliations appears online.

**23andMe Research Team**

Jingchunzi Shi[138] & Suyash S. Shringarpure[138]

**The Biobank Japan Project**

Masato Akiyama[5,6], Yoichiro Kamatani[5,94], Nana Matoba[5,113] & Yukinori Okada[119,120,121,122]

## Methods

Here we describe an overview of the methods used to conduct the association, fine-mapping and downstream in silico functional analysis. Additional details can be found in the Supplementary Note.

### Generation of summary statistics and ancestry considerations

Except for TOPMed studies, in which the genetic data were derived from deep whole-genome sequencing, participants in all studies were genotyped on genome-wide arrays. The majority of studies imputed their genotypes to the Haplotype Reference Consortium[41] (for EUR ancestries) or 1000 Genomes[42] (Supplementary Table 1). GWAS summary statistics were generated in each study sample typically using RVTESTS[43], BOLT-LMM[44] or SAIGE[45] with covariates of sex, age, age squared and genetic principal components according to an analysis plan detailed in the Supplementary Note. Studies composed primarily of closely related individuals (for example, family studies) first regressed out covariates, inverse-normalized the residuals as necessary and then tested additive variant effects under a linear mixed model with a genetic kinship matrix for all phenotypes. Some studies of unrelated individuals followed the same analysis for quasi-continuous phenotypes (AgeSmk, CigDay and DrnkWk), but estimated additive genetic effects under a logistic model for binary phenotypes (SmkInit and SmkCes).

We used terminology and acronyms from the 1000 Genomes Project[42] to describe ancestry. The majority of participating cohorts stratified their sample by ancestry before generation of summary statistics. Cohorts composed of substantial samples of multiple ancestry groups provided summary statistics stratified by ancestry, as well as results based on all individuals regardless of ancestry for use in the multi-ancestry meta-analyses. As TOPMed served multiple functions in the present study, including as an LD reference panel, we detailed the ancestry analyses and classification of TOPMed data in the Supplementary Note. For example, for both ancestry-stratified and multi-ancestry conditional analysis, we created TOPMed reference panels for estimating LD. We first created ancestry-stratified reference samples, resulting in matched ancestry reference sample sizes of $n = 28,665$ AFR, $n = 19,737$ AMR, $n = 4,918$ EAS and $n = 51,656$ EUR. To create a TOPMed-based reference sample for multi-ancestry analyses, we combined the matched ancestry individuals, resulting in a diverse ancestry reference panel ($n = 104,976$) that matches the ancestry proportions of the included cohorts to estimate LD.

Extensive quality control and filtering were performed on the summary statistics from each cohort. We removed studies with a sample size of less than 100, and those with genomic control values greater than 1.1 or less than 0.9 and a sample size of less than 10,000 (per study sample size and genomic control values are listed in Supplementary Table 1), as well as variants with an imputation quality of less than 0.3.

### Ancestry-stratified meta-analyses

Ancestry-stratified meta analyses were performed using the software package rareGWAMA (see URLs for software use). Specifically, the method aggregated weighted $Z$-score statistics, that is,

$$Z_{\text{META}} = \frac{\sum_k w_k Z_k}{\left(\sum_k w_k^2\right)^{1/2}},$$

where $Z_k$ is the $Z$-score statistic in study $k$. The weight $w_k$ is defined by $w_k = \sqrt{N_k p_k (1 - p_k) R_k^2}$, where $p_k$ is the variant allele frequency, and $R_k^2$ is the imputation quality in study $k$. This method accounts for between-study heterogeneity in phenotype measures, imputation accuracy, allele frequencies and sample sizes.

### Multi-ancestry meta-analyses

Multi-ancestry meta-analyses were performed using mixed-effects meta-regression for optimal trans-ancestry meta-analysis (MEMO)

implemented in rareGWAMA (see URLs for software use). The full model is $b_{jk} = \sum_{l=0}^{L} C_{lk} \gamma_{jl} + e_{jk} + \epsilon_{jk}$, where $b_{jk}$ is the genetic effect estimate for the $j$th variant in the $k$th study, and $C_{lk}$ is the $l$th ancestry component for the $k$th study. Note that we set $C_{0k} = 1$, so $\gamma_{j0}$ serves as the intercept. The regression coefficient $\gamma_{jl}$ captures the effect of the $l$th axis of genetic variation for the $j$th variant, with $\gamma_{j0}$ as an intercept in the model, and $e_{jk} \sim N(0, \tau^2)$ is the random effect that captures unexplained effect size heterogeneity after adjusting for genetic variation. Finally, $\epsilon_{jk} \sim N(0, s_{jk}^2)$ is the random error term, where $s_{jk}^2$ is the variance of the genetic effect estimate $b_{jk}$. This method models heterogeneity of effects attributable to ancestry as well as a random effect to capture residual heterogeneity. The MEMO model contains fixed-effect, random-effect and meta-regression models as special cases. Specifically, removing the random effect $e_{jk}$ results in a regular meta-regression model, removing the covariates of genetic variation ($C_{lk}$), but retaining $e_{jk}$ results in a random-effect meta-analysis model, whereas removing both $e_{jk}$ and $C_{lk}$ results in a fixed-effect meta-analysis model.

Per study ancestry variation, $C_{lk}$ is calculated using MDS on the basis of allele frequency. We defined the genetic distance between two studies, that is, study $k$ and $k'$, with $J$ variants, as $d_{kk'} = \sqrt{\sum_j (f_{jk} - f_{jk'})^2}$, where $f_{jk}$ and $f_{jk'}$ are the allele frequency for the $j$th variant for study $k$ and $k'$, respectively. We fit models with 0, 1, 2, 3 and 4 MDS components and combined the results using a minimal $P$ value approach (see Extended Data Fig. 1a for a visual representation of the first four MDS components).

To better ensure robustness, for each phenotype, we filtered variants from the meta-analytic results to variants that were present in at least three studies, had an effective sample size (sample size multiplied by imputation accuracy) to maximum sample size ratio of $\geq 0.1$, and minor allele frequency (MAF) > 0.001 in the multi-ancestry and EUR-stratified meta-analysis or MAF > 0.01 for AMR-stratified, AFR-stratified and EAS-stratified meta-analysis, given the expected drop off in imputation accuracy for those ancestries. These filters reduce potential artefacts arising from sparse data or poor imputation and retain variants with reasonable statistical power.

With increasing imputation accuracy and the inclusion of variants with MAF down to 0.1% (for EUR), genome-wide significant variants were identified using a threshold of $P < 5 \times 10^{-9}$, to account for approximately 10 million independent tests. The threshold was chosen based on previous work on low-frequency variants[5,46,47]. All statistical tests are two-sided unless otherwise stated.

### Robustness and replicability of signals

We applied genomic control correction for low-frequency variants (MAF < 1%) in both multi-ancestry and ancestry-stratified meta-analyses. Genomic control correction for common variants was not applied given that elevation of genomic control values is expected with high polygenicity (that is, it assumes sparsity) and very large sample sizes[48]; such a correction may be overly conservative. To evaluate this decision, we estimated the replicability of associated loci using a trans-ancestry extension of an existing method[6]. This method, 'RATES', incorporates cohort-level summary statistics (single-nucleotide polymorphism (SNP) effect sizes and their corresponding standard errors), along with allele frequency-based MDS components per study to assign a posterior probability that each sentinel variant effect would replicate in a sufficiently powered study. To further evaluate robustness of our results, we estimated LD score regression (LDSC) intercepts and attenuation ratios to account for bias in the intercept test when sample sizes become extreme, as in the present case. Results were within expected limits and consistent with a limited effect of population stratification on the meta-analysis results[44] (Supplementary Table 8). Then, we compared the sign of SNP effect size estimates between EUR-stratified results and within-sibling GWAS results from the UK Biobank, finding sign concordance estimates of 63.4–80% across phenotypes, all of which were significantly higher than would be expected if our results

were driven entirely by population stratification or cryptic relatedness and were consistent in magnitude with other large-scale association studies[49]. Finally, given reduced power in the within-sibling GWAS, we additionally compared the sign of SNP effect size estimates between EUR-stratified 23andMe summary statistics (the largest participating cohort) and EUR-stratified summary statistics with all cohorts except 23andMe, finding sign concordance estimates of 94.3–100%. See the Supplementary Note for further details on the methods and full results, including the list of excluded variants and loci.

## Conditional analyses and locus definitions

We performed sequential forward selection to identify independently associated variants in each locus[50] for ancestry-stratified and multi-ancestry results. The procedure begins by including only the top association signal into a set of independently associated variants ($\phi$) per locus. Conditional analysis is then conducted on the remaining variants, conditioning on variants in $\phi$. If any of these conditional signals remained significant (that is, $P < 5 \times 10^{-9}$), we added the top signal to the set $\phi$. The process iterates until there are no remaining significantly associated variants. The method requires an external genomic reference panel to estimate LD patterns. For ancestry-stratified conditional analyses, we used ancestry-matched individuals from TOPMed to estimate LD (sample sizes given previously). For multi-ancestry conditional analyses, we used the diverse ancestry TOPMed reference panel ($n = 104,976$) that matched the ancestry proportions of the included cohorts.

Loci were defined based in part on the conditional analysis, using a multi-step approach. First, consistent with previous GWAS meta-analysis[5] in EUR ancestries, we identified all 1-Mb windows surrounding sentinel variants and collapsed overlapping windows. This resulted in a total of 1,449 such windows. For each window, we then used our ancestry-aware conditional analysis[51] (described previously) with an ancestry-matched reference panel from TOPMed to enumerate all independent variants within each window. Then, for each independent variant, we defined a locus as the region including all variants in LD of $r^2 > 0.1$, based on the same ancestry-matched TOPMed reference panel (Supplementary Table 3 and Supplementary Fig. 3). Overlapping loci were then collapsed. This procedure avoids conventional definitions of a locus based on work in EUR ancestries and is tailored to the multi-ancestry data at hand.

## Allelic effect size moderation

We evaluated evidence of effect size moderation by ancestry in the multi-ancestry model for each independent variant. To do so, we extended the MEMO model into a mixture model that separated variants with homogenous effects (models with only an intercept term) from those with possible heterogeneous effects (on at least one axis of genetic variation). We considered six sub-models including the null model, and the models in which the number of included components varied from 0 to 4.

$$L(y) = \prod_a p_a^{NULL} p(b_j | NULL) + p_a^{ALT} \sum_{j \in S_a} [q_{j0}\, p(b_j | MR_0(j))$$
$$+ ... + q_{j4}\, p(b_j | MR_4(j))],$$

where $p(b_j | NULL)$ and $p(b_j | MR_l)$ are the likelihoods of the variant $j$ effect sizes under the null model and the meta-regression models with $l$ axes of genetic variation, respectively; $p_a^{NULL}$ and $p_a^{ALT}$ are the probabilities of locus $a$ carrying zero or at least one causal variant, respectively. The term $q_{jl}$ is the probability that the model with $l$ axes of genetic variation best fit the data. We selected the model with the largest posterior probability for each variant as the best-fitting model to capture the genetic effect heterogeneity. Variants in which the zero component model was selected (that is, all models with at least one component were rejected) were considered to have homogeneous effects across

ancestry. Among the remaining variants, we considered which one of the meta-regression models (that is, 1–4 components) best described the extent of effect heterogeneities based on the posterior probabilities for each model. In addition, we required that strongly heterogeneous variants had an MDS component effect that was significantly different from zero and were polymorphic in two or more ancestry-stratified cohorts to ease interpretation of heterogeneous effects. For example, a variant in which the model with two components best fit the data was considered at least weakly heterogeneous. If this variant also had a component two effect significantly different than zero ($\gamma_{j2} \neq 0$, from above) and was polymorphic in at least two ancestries, it was considered strongly heterogeneous.

## Fine-mapping

On the basis of the selected genetic effect model (above), for each variant in a locus, we calculated the Bayes factor by $\Lambda_j = \exp\left[\frac{X_j - (T+1)\log K}{2}\right]$, where $X_j$ denotes the chi-squared test statistic for variant $j$, $T$ denotes the number of axes of genetic variation included in the best-fitting model (that is, 0–4 MDS components) and $K$ denotes the number of studies contributing to the GWAS. Using the approximate Bayes factor, we then calculated the posterior inclusion probability for each variant as $\pi_j = \frac{\Lambda_j}{\sum_i \Lambda_i}$, where $i$ indexes each locus. Finally, we derived 90% credible intervals by ranking variants within a locus by their single posterior estimate and selecting variants until the cumulative posterior inclusion probability reached 0.90.

For EUR-stratified fine-mapping, we approximated the Bayes factor as above with $T$ set to 0. Fine-mapping was conducted in EUR-stratified results, using identical loci as in multi-ancestry fine-mapping, to describe the increased resolution attributable to diverse ancestry inclusion and differences in sample size.

Functional enrichment analysis was conducted to test whether high-priority genes identified in the fine-mapping results were expressed in specific tissue types or enriched in certain cell types or gene pathways. High-priority genes were defined as those mapped from variants in credible intervals containing less than five variants. That is, for each variant in credible intervals with less than five variants, we used the UCSC genome annotation database to assign genes. We assigned intergenic variants to the nearest gene. We mapped genes from variants with PIP < 0.01 (as 'control' genes) in the same way. Functional enrichment was then evaluated by estimating a relative risk (as described and implemented previously[52]), defined as the ratio of the proportion of genes mapped from variants in credible intervals with less than five variants that are in a given annotation category to the proportion of genes mapped from variants, within associated loci, with PIP < 0.01 in the same annotation category. Annotation categories were derived from GTEx tissue expression[53], central nervous systems cell types[50] and gene pathways[54].

## TWAS

TWAS were performed using a trans-ancestry method. In brief, this method fits a series of meta-regression models including the first four axes of genetic variation (MDS components), similar to that of our multi-ancestry meta-analysis model minus the random-effect term. Genetic effect estimates from these four models were then used to estimate phenotypic effects of each variant. Together, with variant weights taken from PrediXcan[55] based on 49 tissues from GTEx[10] release version 8 (which includes up to 15% of individuals of non-EUR ancestry), the phenotypic effect estimates were used to construct a single TWAS statistic for each MDS component. A minimum $P$ value approach[56] was then applied to combine all four TWAS statistic $P$ values. Finally, we used a Cauchy combination test[57] to combine $P$ values across all available tissues for each gene. The final, combined $P$ value was subjected to a Bonferroni correction for 22,121 genes in 49 tissues. We present our TWAS results based on per gene $P$ values combined across all available tissues, resulting in a 5 (phenotype) × 22,121 (gene) matrix of $P$ values.

Pathway enrichment was also conducted using a weighted regression approach[58] with the TWAS per-tissue *P* values to quantify the enrichment of identified genes in each pathway.

## Heritability and genetic correlations

LDSC[59] was used to estimate heritability of our five phenotypes for EAS and EUR ancestries using a standard 1-cM window size. For ancestries with more recent admixture (AFR and AMR ancestries), we used covariate-adjusted LDSC[60] for the same analyses in which in-sample LD scores were calculated using ancestry-matched TOPMed reference samples and adjusted by the first 50 principal components. For more recently admixed AFR and AMR ancestries, which tend to show longer-range LD, we used a 20-cM window size when calculating LD scores. For both LDSC and covariate-adjusted LDSC, variants were subset to HapMap3 (ref. [61]) with MAF > 0.05, as recommended for this approach.

We calculated genetic correlations between our five phenotypes and 4,065 UK Biobank phenotypes (both restricted to EUR ancestry) using bivariate LDSC with 1000 Genomes-based pre-calculated EUR LD scores for HapMap3 variants. We excluded phenotypes with heritability *Z*-scores less than 3 (reflecting near-zero heritability), genetic correlations with our phenotypes less than −0.8 or greater than 0.8, to remove phenotypes approaching redundancy with our target tobacco and alcohol use measures (for example, cigarettes per day versus packs per day), and those whose genetic correlations were unable to be estimated largely due to negative heritability estimates, leaving 1,141 UK Biobank phenotypes. Affinity propagation clustering[62], a message-passing algorithm based on exemplars that identifies their corresponding set of clusters, was then used to further interpret the pattern of genetic correlations and multifactorial nature of substance use. A Bonferroni-corrected *P* value threshold for 1,141 UK Biobank phenotypes was used to identify genetic correlations that were significantly different from zero.

## Polygenic scoring

Polygenic risk scores were computed using LDpred for each ancestry group separately, an approach that incorporates the correlation between genetic variants to re-weight effect size estimates[63]. We used an independent prediction cohort, Add Health[35], to validate each score. Add Health is a nationally representative sample of US adolescents enrolled in grades 7 through 12 during the 1994–1995 school year. The mean birth year of respondents was 1979 (s.d. = 1.8) and the mean age at assessment (here, wave 4) was 29.0 years (s.d. = 1.8), which is comparable, in general, to the age of participants in the 23andMe cohort but younger, on average, than those in other cohorts. Add Health is composed of individuals from the same four major ancestral groups (defined with reference to 1000 Genomes; see Supplementary Note for details) comprising our ancestry-stratified results (EUR, AFR, AMR and EAS). Phenotypic descriptive statistics are given in Supplementary Table 12. Across the full Add Health sample, approximately 41% ever smoke regularly and reported an average of 7.3 cigarettes per day. For each polygenic score, we used only HapMap3 variants and those with MAF > 0.01. We used each Add Health ancestry group as its own LD reference panel for construction of each polygenic score, after removing related individuals, except for EAS in which we use 1000 Genomes due to the small sample size in Add Health.

Prediction accuracy of each polygenic score was estimated by taking the difference in the coefficient of determination ($R^2$) between a base model that included only the covariates of age, sex, age × sex interaction, and the first ten genetic principal components, and a full model that additionally included the polygenic score. All scores were scaled to have a mean of zero and standard deviation of one.

## URLs for software use

BCFtools, http://samtools.github.io/bcftools/; BOLT-LMM, https://data.broadinstitute.org/alkesgroup/BOLT-LMM/; cov-LDSC, https://github.com/immunogenomics/cov-ldsc; EAGLE, https://alkesgroup.broadinstitute.org/Eagle/; GCTA, http://cnsgenomics.com/software/gcta/; IMPUTE2, https://mathgen.stats.ox.ac.uk/impute/impute_v2.html; LDpred, https://github.com/bvilhjal/ldpred/; LDSC, https://github.com/bulik/ldsc/; MEMO (rareGWAMA), https://github.com/dajiangliu/rareGWAMA/; Minimac3, https://genome.sph.umich.edu/wiki/Minimac3; PLINK, https://www.cog-genomics.org/plink/; R, https://www.r-project.org/; RATES, https://github.com/wangc29/RATES; RVTESTS, https://github.com/zhanxw/rvtests/; SAIGE, https://github.com/weizhouUMICH/SAIGE; SHAPEIT, http://mathgen.stats.ox.ac.uk/genetics_software/shapeit/shapeit.html; TESLA, https://github.com/funfunchen/rareGWAMA; VCFtools, https://vcftools.github.io/index.html.

## Ethics

Ethical review and approval were provided by the University of Minnesota institutional review board. All human participants provided informed consent.

## Reporting summary

Further information on research design is available in the Nature Portfolio Reporting Summary linked to this article.

## Data availability

GWAS summary statistics can be downloaded online (https://doi.org/10.13020/przg-dp88) with more information available here: https://genome.psych.umn.edu/index.php/GSCAN. We have provided association results for variants that passed quality-control filters in the multi-ancestry and ancestry-stratified results for each of the five substance use phenotypes, excluding data provided by 23andMe. Ancestry-stratified polygenic score weights based on ancestry-stratified summary statistics are also provided. 23andMe results are available directly from the company.

## Code availability

All software used to perform these analyses is publicly available. Software tools used are listed in the main text and Methods.

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

**Acknowledgements** This study was designed and carried out by the GWAS and Sequencing Consortium of Alcohol and Nicotine use (GSCAN). It was conducted by using the UK Biobank Resource under application number 16651. This study was supported by funding from US National Institutes of Health awards R56HG011035, R01DA044283, R01DA042755 and U01DA041120 to S.V., and R01GM126479, R56HG011035, R03OD032630, R01HG011035 and R56HG012358 to D.J.L. G.R.B.S. was also supported by National Institutes of Health award T32DA050560. D.J.L. and X.W. were in part supported by the Penn State College of Medicine's Biomedical Informatics and Artificial Intelligence Program in the Strategic Plan. A full list of acknowledgements is provided in the Supplementary Note.

**Author contributions** D.J.L. and S.V. designed, led and oversaw the study. G.R.B.S and X.W. were the lead analysts for the study, and they were assisted by D.J.L., S.V., F. Chen, S.-K.J., M. Liu and C.W. Phenotype definitions were developed by L.J.B., M.C.C., J. Kaprio., E.J., D.J.L., M. McGue, M.R.M., S.V. and L.Z. Software development was carried out by X.W., D.J.L., F. Chen and C.W. Multi-ancestry meta-analyses were performed by X.W. Ancestry-stratified meta-analyses were performed by G.R.B.S. and X.W. Meta-analyses were performed by X.W. and G.R.B.S. Fine-mapping and allelic heterogeneity were performed by X.W. and G.R.B.S. Replicability analyses were performed by C.W., S.-K.J. and G.R.B.S. Multi-ancestry TWAS were performed by F. Chen. Heritability and genetic correlation analyses were performed by S.-K.J. Polygenic scoring analyses was performed by G.R.B.S. Bioinformatics analyses were performed and interpreted by F. Chen, S.-K.J., G.R.B.S., S.V. and J.A. Stitzel. Figures were created by M. Liu, G.R.B.S., S.-K.J. and S.V. M. Liu and S.V. coordinated among participating cohorts. M.A.E. and M.C.K. helped with data access. G.R.B.S. coordinated authorship and acknowledgement details. C.B., A.W.B., L.B., S.P.D., S.A.G.T., D.B.H., M.R.M. and T.E.T. provided helpful advice and feedback on study design and the manuscript. All authors contributed to and critically reviewed the manuscript. G.R.B.S., X.W., S.-K.J., F. Chen, C.W., D.J.L. and S.V. made major contributions to the writing and editing.

**Competing interests** The spouse of N.L. Saccone is listed as an inventor on issued U.S. patent 8080371 'Markers of addiction', covering the use of certain single-nucleotide polymorphisms in determining the diagnosis, prognosis and treatment of addiction. M.H.C. has received grant funding from GSK and Bayer, and speaking or consulting fees from AstraZeneca, Illumina and Genentech. R.T.-S. is a former employee and current shareholder of GSK and is currently a non-executive member of the ENA Respiratory board of directors. She reports personal fees from Teva, Immunomet, Vocalis Health and ENA Respiratory (until January 2021). D.A.S. is the founder and chief scientific officer of Eleven P15, a company focused on the early diagnosis of treatment of pulmonary fibrosis. J.B.N. and E.J. are employed by Regeneron Pharmaceuticals, Inc. The spouse of C.J.W. is employed by Regeneron Pharmaceuticals, Inc. L.J.B. is listed as an inventor on Issued U.S. Patent 8080371 'Markers for addiction', covering the use of certain single-nucleotide polymorphisms in determining the diagnosis, prognosis and treatment of addiction. The 23andMe Research Team, including J.S. and S.S.S., are employees of 23andMe, Inc., and hold stock and/or stock options in 23andMe. T.E.T., D.F.G., H.S., G.B. and K. Stefansson are employees of deCODE genetics/AMGEN. M. Moll received grant support from Bayer. A.W.B. is listed as a co-inventor on a U.S. patent application 'Biosignature discovery for substance use disorder using statistical learning' assigned to BioRealm, LLC, and serves as a scientific advisor and consultant to BioRealm, LLC. All other authors declare no competing interests.

**Additional information**
**Correspondence and requests for materials** should be addressed to Dajiang J. Liu or Scott Vrieze.

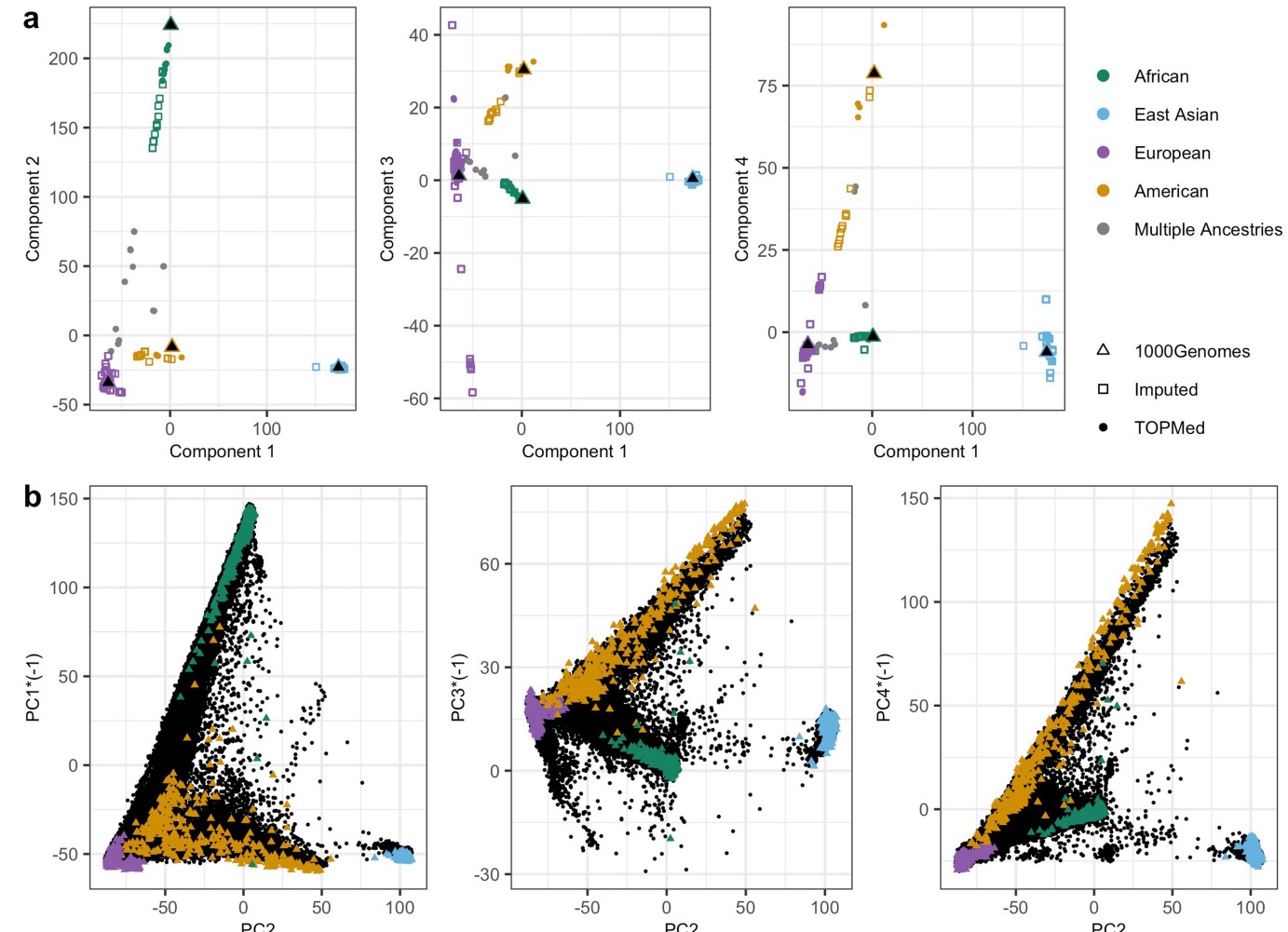

**Extended Data Fig. 1 | Ancestry space of studies contributing to meta-analysis (panel a), versus individuals from TOPMed and 1000 Genomes (panel b).** The meta-regression within the MEMO model requires specification of ancestry clines. To ensure consistency in the meaning of ancestry clines across all five MEMO analyses (one for each phenotype) we created a single multidimensional scaling solution based on allele frequencies from all phenotypes in all participating cohorts. These solutions are plotted in panel a (circles correspond to TOPMed cohorts, squares are all other cohorts which used imputed microarray genotypes, and triangles are 1000 Genomes ancestry groups). Colors of points correspond to the primary assigned ancestry of each cohort (studies with < 90% of individuals coming from a single ancestry group are shown in grey). Panel b shows projection of principal components (after OADP transformation) of TOPMed individuals onto PCs of 1000 Genomes individuals, in colored triangles. Each 1000 Genomes individual is colored by their known ancestry. This PC information was used in assigning ancestry to TOPMed individuals for the purpose of reference panel creation (individuals of South Asian ancestry were not included in analyses). The PCs in panel b were reordered or reversed in some cases to align with panel a. These transformations are noted in the axis labels.

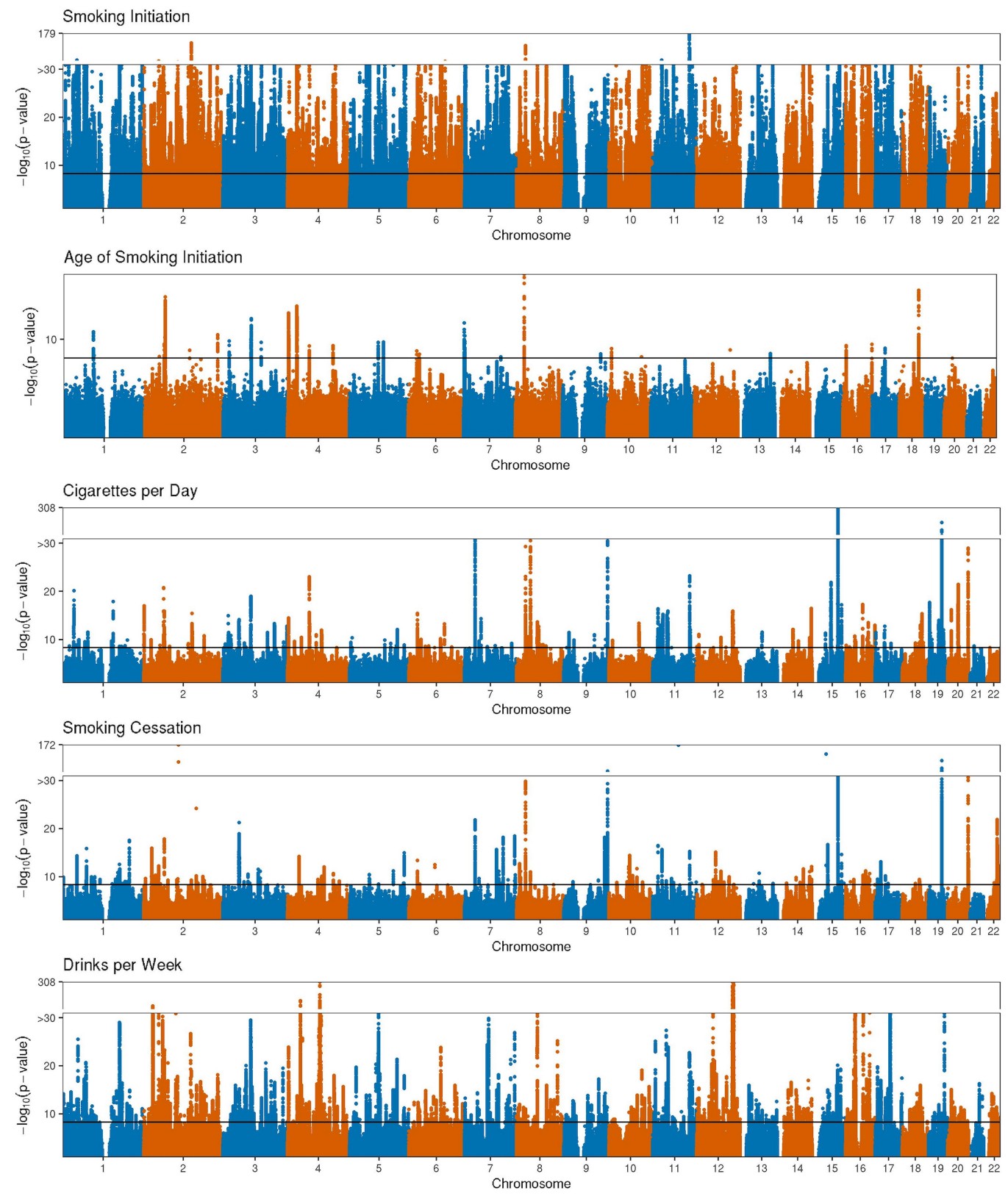

**Extended Data Fig. 2 | Multi-ancestry meta-analysis Manhattan plots.** Black horizontal line corresponds to $P = 5 \times 10^{-9}$, the GWAS significance threshold used for all analyses. Note that some y-axis scales are discontinuous to better illustrate variants with very small $P$-values (e.g., the Drinks per Week y-axis is cut at 30 with a maximum value of 307.7, denoting a $P$-value of $1.9 \times 10^{-308}$). All $P$-values are from two-sided statistical tests.

## a Tissue Expression Enrichment

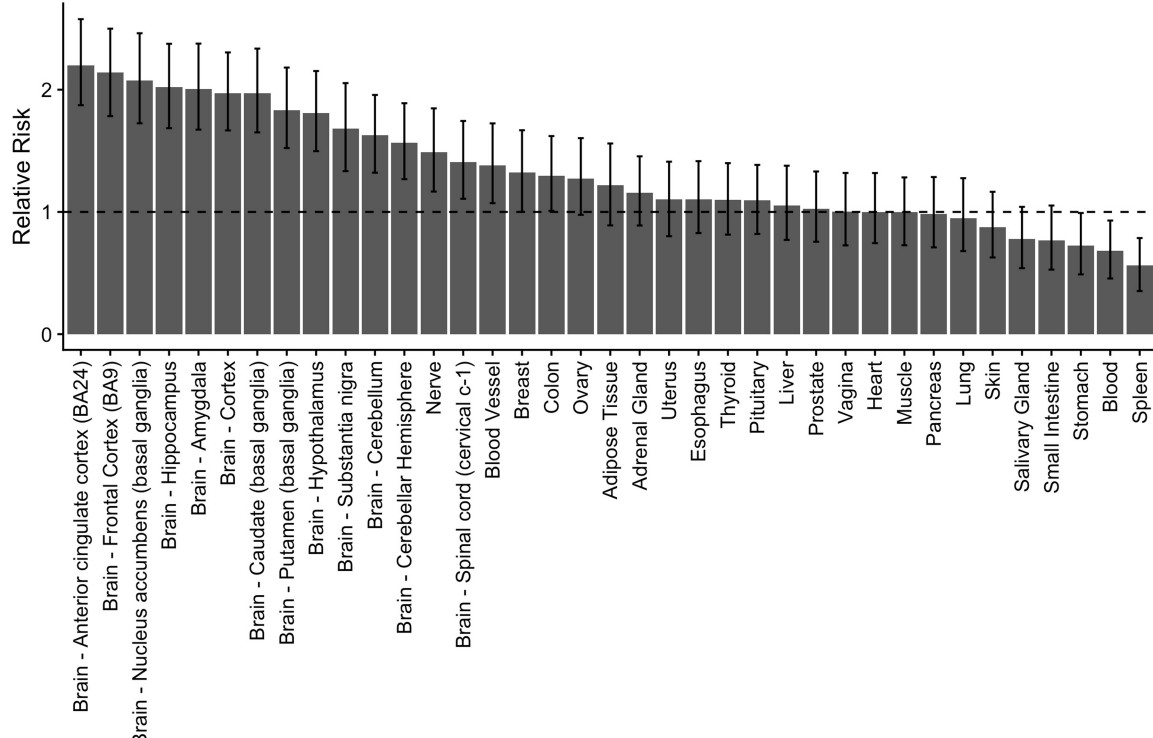

## b Brain Cell Type Enrichment

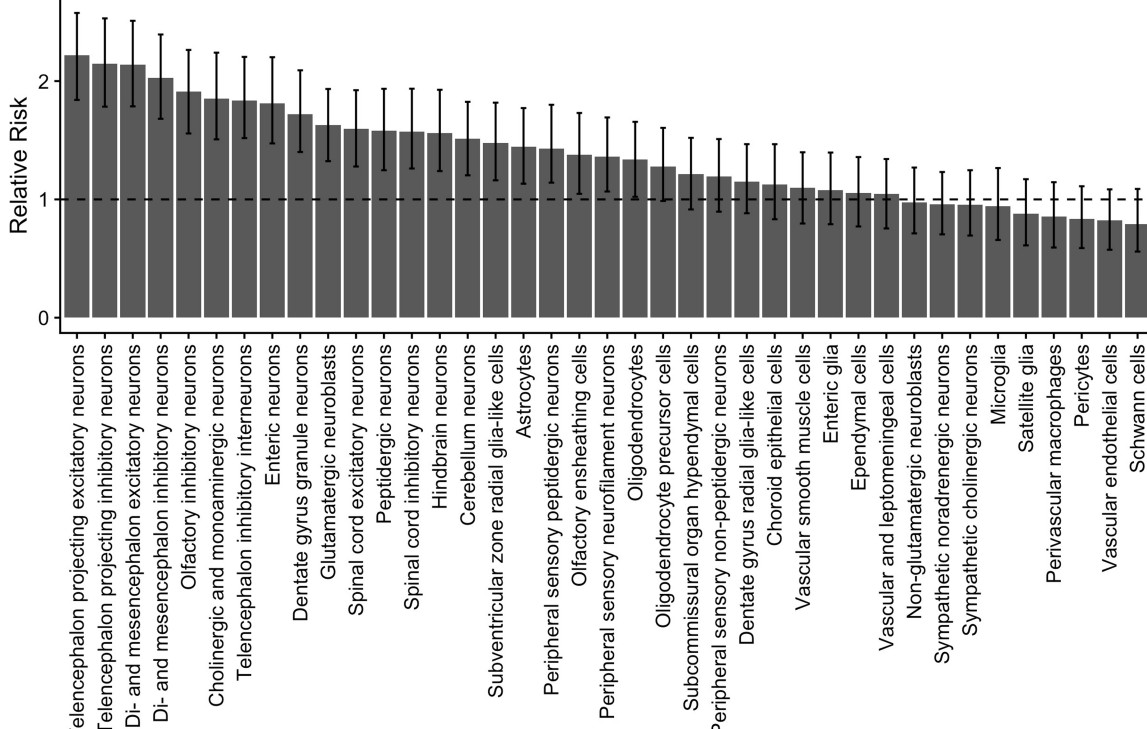

**Extended Data Fig. 3 | Tissue expression and brain cell type enrichment in high priority genes.** Panel a shows tissue expression enrichment in 'high priority' genes. We define high priority genes here as those located nearest to the variants in fine-mapped credible intervals containing less than five variants. These genes were compared to 'control' genes identified in the same way, but from variants in credible intervals with PIP < 0.01 from the trans-ancestry fine-mapping. The x-axis denotes GTEx tissue types. The y-axis represents relative risk estimates comparing high priority to control genes. Panel b shows similar relative risk comparisons with 39 brain cell types. Data are presented as relative risk values with error bars denoting bootstrapped 95% confidence intervals. Further details on estimating relative risk are included in the Supplementary Note section 'Functional enrichment'.

Scott Vrieze

# Reporting Summary

## Statistics

For all statistical analyses, confirm that the following items are present in the figure legend, table legend, main text, or Methods section.

| n/a | Confirmed | |
|---|---|---|
| ☐ | ☒ | The exact sample size (*n*) for each experimental group/condition, given as a discrete number and unit of measurement |
| ☐ | ☒ | A statement on whether measurements were taken from distinct samples or whether the same sample was measured repeatedly |
| ☐ | ☒ | The statistical test(s) used AND whether they are one- or two-sided *Only common tests should be described solely by name; describe more complex techniques in the Methods section.* |
| ☐ | ☒ | A description of all covariates tested |
| ☐ | ☒ | A description of any assumptions or corrections, such as tests of normality and adjustment for multiple comparisons |
| ☐ | ☒ | A full description of the statistical parameters including central tendency (e.g. means) or other basic estimates (e.g. regression coefficient) AND variation (e.g. standard deviation) or associated estimates of uncertainty (e.g. confidence intervals) |
| ☐ | ☒ | For null hypothesis testing, the test statistic (e.g. *F*, *t*, *r*) with confidence intervals, effect sizes, degrees of freedom and *P* value noted *Give P values as exact values whenever suitable.* |
| ☐ | ☒ | For Bayesian analysis, information on the choice of priors and Markov chain Monte Carlo settings |
| ☒ | ☐ | For hierarchical and complex designs, identification of the appropriate level for tests and full reporting of outcomes |
| ☐ | ☒ | Estimates of effect sizes (e.g. Cohen's *d*, Pearson's *r*), indicating how they were calculated |

*Our web collection on statistics for biologists contains articles on many of the points above.*

## Software and code

Policy information about availability of computer code

| Data collection | No software was used. |
|---|---|
| Data analysis | All studies used either ShapeIt2 or EAGLEv2.4 to phase genotypes and used either Minimac3 or IMPUTE2 for imputation. Summary statistics were generated using RVTESTS release v1.9.7 or v1.9.9, BOLT-LMM v2.3.2, or SAIGE v0.35.8.1. Standard quality control used PLINK v1.9 or v2.0, BCFtools v1.9, and VCFtools v0.1.16. Meta-analyses and conditional analyses were performed using rareGWAMA v0.7 in R v3.6.0 and GCTA v1.93.0 and v1.94.0. Transcriptome-wide association analysis was performed using TESLA (implemented in rareGWAMA). Replicability of associated loci was performed using RATES v1.0.0. Standard and covariate-adjusted LD Score Regression was used to measure heritability, test for population stratification, and estimate genetic correlations (LDSC v1.0.1 and cov-LDSC v1.0.0). LDpred v1.0.8 was used to construct the polygenic scores. |

For manuscripts utilizing custom algorithms or software that are central to the research but not yet described in published literature, software must be made available to editors and reviewers. We strongly encourage code deposition in a community repository (e.g. GitHub). See the Nature Portfolio guidelines for submitting code & software for further information.

## Data

Policy information about availability of data

All manuscripts must include a data availability statement. This statement should provide the following information, where applicable:
- Accession codes, unique identifiers, or web links for publicly available datasets
- A description of any restrictions on data availability
- For clinical datasets or third party data, please ensure that the statement adheres to our policy

GWAS summary statistics can be downloaded online (https://doi.org/10.13020/przg-dp88) with more information available here: https://genome.psych.umn.edu/

index.php/GSCAN. We provide association results for variants that passed quality-control filters in the multi-ancestry and ancestry-stratified results for each of the five substance use phenotypes, excluding data provided by 23andMe. Ancestry-stratified polygenic score weights based on ancestry-stratified summary statistics are also provided. 23andMe results are available directly from the company.

# Field-specific reporting

Please select the one below that is the best fit for your research. If you are not sure, read the appropriate sections before making your selection.

☒ Life sciences   ☐ Behavioural & social sciences   ☐ Ecological, evolutionary & environmental sciences

For a reference copy of the document with all sections, see nature.com/documents/nr-reporting-summary-flat.pdf

# Life sciences study design

All studies must disclose on these points even when the disclosure is negative.

| | |
|---|---|
| Sample size | No sample size calculation was necessary as we increased our sample size to the extent possible. We contacted as many studies (with our phenotypes of interest) as possible and applied for relevant studies available in public repositories. Our meta-analysis includes the largest sample size of similar phenotypes to date and therefore, our results are sufficiently powered. |
| Data exclusions | We excluded results from smaller studies when those results behaved unusually (e.g., inflated or deflated genomic controls), and there was no alternative explanation (e.g., inflation was due to polygenic signal). We applied filters to the genomic data post meta-analysis (minor allele frequency > .1% for European-stratified results or > 1% for all others, effective sample size of at least 1% per phenotype and at least 3 studies must be included for each variant) in order to only report variants on which we had robust results. Finally, we removed 17 loci in which the lead SNPs posterior probability of replicability fell below a threshold of .99, although these are reported in the supplementary materials. |
| Replication | In order to maximize power to detect the variants, we did not separate our sample into a separate discovery and replication set. We used a trans-ancestry extension of the Meta-Analysis Model-based Assessment for replicability (MAMBA) to assess the posterior probability of replicability of associations without an independent replication sample. References are available in the manuscript. |
| Randomization | N/A. No randomization was employed as the current study was observational and used all available participants. |
| Blinding | N/A. Blinding was not applicable to the current study as we did not employ any intervention. |

# Reporting for specific materials, systems and methods

We require information from authors about some types of materials, experimental systems and methods used in many studies. Here, indicate whether each material, system or method listed is relevant to your study. If you are not sure if a list item applies to your research, read the appropriate section before selecting a response.

### Materials & experimental systems

| n/a | Involved in the study |
|---|---|
| ☒ | ☐ Antibodies |
| ☒ | ☐ Eukaryotic cell lines |
| ☒ | ☐ Palaeontology and archaeology |
| ☒ | ☐ Animals and other organisms |
| ☐ | ☒ Human research participants |
| ☒ | ☐ Clinical data |
| ☒ | ☐ Dual use research of concern |

### Methods

| n/a | Involved in the study |
|---|---|
| ☒ | ☐ ChIP-seq |
| ☒ | ☐ Flow cytometry |
| ☒ | ☐ MRI-based neuroimaging |

## Human research participants

Policy information about studies involving human research participants

| | |
|---|---|
| Population characteristics | All participants were adults. We included all available individuals of all genders and sexes from European, African, American, or East Asian ancestry populations. |
| Recruitment | We did not do any recruitment. Analysis was of existing de-identified data. |
| Ethics oversight | University of Minnesota IRB |

Note that full information on the approval of the study protocol must also be provided in the manuscript.

