## [Peer Review File · Nature]

Manuscript Title: Genetic diversity fuels gene discovery for tobacco and alcohol use

Reviewer Comments & Author Rebuttals

Reviewer Reports on the Initial Version:

Referees' comments:

Referee #1 (Remarks to the Author):

Review of "Trans-Ancestry Genome-Wide Investigation of Tobacco and Alcohol Use in up to 3.4 Million Individuals" by Saunders et al.

The authors report results from genome-wide analyses of four smoking phenotypes (smoking initiation, age at smoking initiation, cigarettes per day, smoking cessation) and drinks per week across major clines of recent human ancestry (AFR, AMR, EAS, EUR) involving from 728,826 individuals (AgeSmk) up to 3,383,199 individuals (smoking initiation). Besides trans-ancestral analyses, ancestry specific analyses were conducted primarily with the aim of evaluating how polygenic scores (PGS) derived from weights obtained from ancestry specific GWASs predicted the phenotypes within and across ancestries.

The authors identified 2,143 genome-wide significant loci ($P < 5 \times 10^{-9}$) across all phenotypes, using a mixed effects meta-regression method that accounts for ancestry clines and other heterogeneities. This method demonstrated to have more power to detect loci than traditional fixed effect meta-analysis not accounting for ancestry. When using a fine-mapping method that accounts for ancestry 28% of identified loci included less than 5 credible variants. Fine-mapping was improved when including ancestry information e.g., the number of causal variants decreased on average with 20% within loci.

Overall, a small proportion of conditionally independent variants (4%) demonstrated large differences across ancestries in effect sizes. PGS analyses demonstrated EUR-based PGS in EUR ancestries to perform better than ancestry matched PGS in non-EUR ancestries.

The subject of the study i.e., cross-ancestry genetic analyses and how to improve and generate further information about the genetic architecture of complex disorders in non-EUR ancestries, is very timely and highly relevant both from a scientific and clinical point of view. The study is very well performed, carefully executed with use of the newest methods, and dubious results are addressed thoughtfully in sensitivity/follow-up analyses (e.g., the unusual genetic correlation pattern of DrnkWk driven by the 23andMe cohort). The manuscript is well written, and the results are interesting, especially I find it interesting to see the large improvement in fine-mapping, when including ancestry information.

Several of the methods applied are new to me, but as far as I could evaluate the statistic analyses are well-performed and I have no major comments or concerns regarding those. I have a few other comments which you can find below:

1) Regarding the phenotype "Drinks per week", it has been shown that the genetic correlation between alcohol consumption and alcohol use disorder is around 0.5 (in EUR individuals) which suggest that alcohol use disorder does not represent the extreme end of a continuous distribution of alcohol consumption when considering the underlying genetic architecture (at least in Europeans). I assume some cohorts have a high representation of individuals with alcohol use disorder. Have the authors considered that this might introduce heterogeneity in the Drinks per week phenotype?

2) When defining genes of interest, the authors apply a filtering step where only loci with better

fine-mapping results in the trans-ancestry analysis compared to the EUR-stratified GWAS results were retained. I do not understand why this was required; I assume the aim would be to extract genes that are likely to affect the phenotypes in order to get a better understanding of the underlying biological factors. So why exclude loci which were nicely fine-mapped already (and did not improve further in the trans-ancestry analysis)? I think it would make most sense to focus on all loci with a set of credible variants less than five from the trans-ancestral GWAS MA. Or at least include the 192 loci where the credible set contained only one variant, no matter improvement in fine-mapping in the trans-ancestry GWAS. Additionally, the authors highlight that there was no evidence of effect moderation by ancestry for most conditionally independent variants, so I assume biological knowledge generated by focusing on all loci with few credible variants in the trans-ancestry GWAS would be relevant for all ancestries.

3) Regarding my point above, the authors intersect their GWAS results with TWAS findings when identifying genes of special interest. Even though a huge amount of work has been performed in order to map out regulatory variants, the best predictor for the causal gene is still distance from the index variant to the gene. Therefore, I would recommend the authors to include genes, in their list of high priority genes, if the signal (determined by credible variants) is located within a gene, even though the TWAS results do not highlight the gene.

4) Please list the high priority genes in a supplementary table together with information about the credible variants and their annotations (e.g., coding/non-coding, eQTL information etc.).

5) I do acknowledge that the focus of the study is on how trans-ancestry genetic information can improve locus discovery and PRS analyses, rather than focusing on the biological factors underlying the phenotypes. However, the authors manage to pinpoint 597 loci with less than 5 credible variants, and 192 loci where the credible set contained only one variant. I think this is a very successful fine-mapping which should give less noise in high-priority genes mapped by credible variants and opens for the opportunity to get closer to the biology of the phenotypes. I miss further analyses that combines high priority genes with functional genomics data. Are these genes enriched in specific pathways or expressed in particular cell types that can inform us more about the underlying biological causes driving the phenotypes?

Minor comments

6) In the analyses evaluating heterogeneity in effect sizes across ancestries the authors evaluate conditionally independent variants. I assume they evaluated genome-wide significant variants, but I don't think it is clear from the text, or maybe I missed it somewhere?

7) In the TWAS the authors correct for 22,121 genes, I think it would be appropriate to take the number of tested tissues into consideration when correcting for multiple testing.

8) Did the authors normalize their polygenic scores?

9) What was the reason to include "age2" as a covariate in the PGS analyses?

10) In supplementary table 6, could the authors include a column indicating if the heritability estimates are significantly different from zero.

11) Supplementary Figure 4, B1, the y-axis is named "PC2*(-1)" should that be "PC1*(-1)"?

12) Supplementary table 6, there is an unusual high ratio in the AFR, AgeInit analysis (ratio = 0.609). This indicates that a very high proportion of the inflation in the test statistics is caused by other factors than polygenic heritability. I think this is unusual high, could the authors comment on this?

Referee #2 (Remarks to the Author):

Overview: Saunders, Wang, Chen, Jang, Liu, Wang, et al. present their work investigating the genetic contribution to tobacco and alcohol use phenotypes in a large-scale multi-ancestry analysis. Overall they report numerous genomic risk regions associated with these phenotypes, the majority of which are consistently identified across ancestry groups. In line with prior works, multi-ancestry fine-mapping considerably reduces credible SNP sets, which are used to illustrate several genes with putative effects on smoking phenotypes identified through TWAS. Their work contains numerous statistical advancements, which will have broad applicability to the field of multi-ancestry statistical genetics. The manuscript is well written, and the results presented in a clear, understandable manner. The authors have performed detailed validation when appropriate and I commend them for their overall efforts in combining numerous genetic and genomic datasets. I do have a few minor comments related to some of the claims.

Major Comments:

1. "Some of these variants unique to trans-ancestry results are located in genes related to neurotransmission (e.g., NRXN1) including glutamatergic (GRIN2A) neurotransmitter systems, of relevance to neurocircuitry in addiction^{11,12}."

The authors report that their meta-regression approach improves power to detect risk regions, and provide support for this fact with some risk variants falling near genes related to neurotransmission. What is unclear to me is if this is expected by chance or not, given the large increase in the number of associations found. It would be helpful if the authors could perform some permutation test or enrichment test to provide stronger statistical support for these findings.

2. "There was no evidence of effect moderation by ancestry for most conditionally independent variants, ranging from 79.2% (194 variants) in CigDay, to 89.9% (187 variants) in SmkCes. Another 10.3% to 16% showed modest evidence for allelic heterogeneity. Finally, roughly 4% of all conditionally independent variants, reflecting 137 variants from 86 distinct loci, showed strong evidence of effect size moderated by ancestry (complete results in Supplementary Table 2)."

I was glad to see this analysis performed, which provides strong support for the overall meta-regression and meta-analysis across ancestries. While a small percentage exhibited evidence of heterogeneity, can the authors demonstrate that these are not driven by differential genotyping quality/imputation at these regions. Furthermore, can the authors report F_{st} at these sites compared to the background? Similarly, can the authors report the ancestry-matched LD scores at these sites compared to the background?

Minor Comments:

1. Their cross-ancestry replication method appears to provide support for the overwhelming majority of genomic risk regions, which is consistent with intuition and previous studies, however there were a small number that failed to replicate (17 variants with $\text{prob} < 0.99$). Can the authors provide some additional information on how their model decided that these variants had less evidence of replication.

2. The TESLA method appears to apply eQTL prediction weights to the GWAS effect-sizes adjusted for sampling error (the fitted effects from MEMO), but do not include the heterogeneity term (random effect term in MEMO). This doesn't appear to have a justification, and I was somewhat puzzled given the primary GWAS results are reported using the random-term.

Referee #3 (Remarks to the Author):

Review of "Trans-Ancestry Genome-Wide Investigation of Tobacco and Alcohol Use in up to 3.4 Million Individuals"

This is a follow-up study of the GSCAN study published in 2019 in Nature Genetics, which included 1.2M individuals. The major novelties of the current study are the impressive sample size and the inclusion of diverse ancestries. The authors conduct a GWAS in 3.3M individual and identify thousands of loci linked to tobacco and alcohol consumption. I believe this article provides important new insights into the genetics of alcohol and tobacco consumption and will be of interest to readers. The authors identify The manuscript is well written and the methods are appropriate. The results are presented in a clear and concise manner. Generally, the use of statistics is appropriate and uncertainties well described except for the PRS analysis, I believe (see specific comments below). I have some suggestions for improvement.

1) it seems that no covariates (e.g., gender, SES) were included in the GWAS. Can a statement be included to clarify this, in the methods section of the main text (and potentially also in the results section). I believe this is important information for interpretation.

2) Do any of the MDS components potentially represent social/cultural factors? E.g., there are large cultural differences in drinking behaviors within Europe, so it may be worth investigating potential confounding factors

3) TWAS analyses lead to the identification of many genes. I believe it would be of interest to further explore the biological mechanisms implicated by these genes, e.g., through pathway or gene-set analyses.

4) Can the authors provide additional information on the putative causal variants in the CHRNA5-A3-B4 locus? Did their analysis allow for pinpointing the causal molecular mechanisms that underlie this association? This has been widely debated in the field, and I feel these data provide a unique opportunity to establish the putative causal genes in this locus (this has been difficult previously, due to high LD)

5) Lines 449-450 (comparison of PRS across ancestries). Can more info on statistics be included? I would expect low statistical power to detect differences in predictive accuracy, given the relatively small sample sizes of the target samples. The results in Figure 2 confirm this, SE's are large and often overlap across ancestries. Many SE's (especially for non-European populations) include 0. I'm not sure the results as described in the main text appropriately acknowledge the limited power to identify differences.

6) Lines 468 and further. I believe the interpretation of these findings as being in line with the omnigenic model is a stretch. The authors have not differentiated between core and peripheral genes in their analyses. For this, integration of gene expression networks would be required. I believe that the discussed should be toned down, or additional analyses would need to be performed.

7) Lines 479-481. I don't agree that exposure to tobacco and alcohol will be similar for all study participants, variation in culture and socio-economic status will be important. I do agree that variation may be even more extensive at a global level, so a slight modification of these sentences would probably be sufficient.

8) Can the authors comment on the suitability of their GWAS method in the context of admixed samples. Would it be worth comparing to other recent approaches, e.g., tractor? This is not my area of expertise, but a few sentences on how these methods compare may be helpful

9) Figure 2: I find panel A difficult to read, colors can't be distinguished in the AFR target sample. Also, does the SE of the colored bars represent the SE of all ancestries combined? The error bars seem to overlap across the different ancestry target samples and tend to be very large, often including zero (except for the European target sample). Is the power sufficient to interpret these differences? (related to comment 5)

Minor:

- Lines 367-372, please explain that information is combined across tissues as explained in detail in the methods section
- Supplementary tables: Please provide more extensive explanation, define abbreviations and

define information in columns.

- Line 414: I suggest to add "genetic correlation" to the title of this section

Referee #4 (Remarks to the Author):

This is potentially an important manuscript, and could be appropriate for publication in Nature. I am broadly happy with the analysis methods as I understand them, and I applaud the emphasis on treating ancestry as continuous rather than discrete (though there still remains a substantial amount of analysis in ancestry groups, which is reasonable).

However I am concerned about multiple confusing statements describing methods that indicate lack of understanding and/or care. None of these is individually a major cause for concern but collectively they raise a concern about the authors' grasp of the methods employed, which are often not well explained so that there remain important aspects of the analysis methods that I do not understand.

Some examples of wording that causes concern:

L310 "posterior probability of replication $> .99$ " this doesn't make sense to me, replication probability depends on the replication study and it makes no sense to try to report it. I suspect that what they are reporting is a posterior probability of a non-zero effect that is large enough that a well-powered replication study will detect it, but this requires at least a brief explanation in the main text and more details elsewhere.

L311 This is the first of many instances of the phrase "conditionally independent variants" which doesn't make sense to me. Conditional independence involves three variables: A and B may be unconditionally dependent but independent conditional on C. I suspect that the authors just mean "independent", not "conditionally independent", as independence is often checked by a conditioning analysis: if A and B are independent causal variants, then the effect size of A is not affected by conditioning on B and vice versa.

L316 "high expected replicability" similar to L310, I don't understand this. Once again I think the authors may mean a high posterior probability of an effect size above some threshold.

L325 "2,186 variants that were identified only by ... illustrating an increase in power." How many variants were only identified by the alternative analysis? No mention seems to imply zero but I am sceptical about this, and if non-zero then not reporting it is an omission that could be seriously misleading. Also increased hits do not prove an increase in power, better to be restrained and not claim this - it isn't needed.

L331 "ancestral variation in LD" another confusing phrase, possibly what is meant is variation in LD across ancestry groups? We can't measure LD in ancestral pops.

L332 "90% credible intervals of the posterior probability of inclusion" Another worrying misuse of terms suggesting poor understanding: a probability interval for a probability doesn't make much sense. I guess from context that what is meant is a credible interval for the location of a causal variant.

L334 "a median of 9–19 variants" a median is a single number. As you present a range, I guess you are reporting the median for several datasets, presumably one for each ancestry group but this needs clarity.

L346 this is the first of many uses of the word "moderated" without clarification of what that

means, which can be gleaned from methods but it isn't straightforward. In effect I think "moderated by ancestry" means that the effect size differs significantly across ancestry groups.

L363 "take two people who both carry one copy of the protective T allele for this variant but are separated by one standard deviation on MDS component 1 ... **despite the same genotype.**" Confusing wording because the large difference on the MDS component means that they have many differences in genotype. Omit the last 4 words and the sentence is fine.

L435 "highlighting their relevance to adverse disease outcomes" seems to misunderstand genetic correlations. The high correlation says that the smoking variables and the disease outcomes have to a large extent the same genetic causes, which in fact diminishes their "relevance", suggesting that smoking variables are not directly causal for disease but in fact both are caused by (largely) the same genetic mechanisms.

L444 ".. over and above scores already entered into the model." I do not understand this, are we talking about polygenic scores (if so how do they differ from the new ones) or do you mean non-polygenic regression predictors (in which case best not call them scores). I also do not understand "EUR validation ancestries" here (L446), eventually it is explained in methods but not clear here.

L470 "core disease-associated pathways (e.g., those that are genome-wide significant)" this seems to misunderstand the concept of core pathways, statistical significance is not relevant to understanding the concept.

L546 More poor wording "inflation of test statistics is expected with high polygenicity" if there are many large test statistics due to polygenicity, this is not "inflation" which refers to spuriously large values e.g. due to confounding. Better just to say that GC assumes sparsity which is generally inappropriate but may be OK for low-MAF variants

L928 (Fig 2) The caption for Panel (c) is incomprehensible to me. I don't understand the model description and the "for example" doesn't seem to relate to the model description that precedes it

Other issues:

L320 "Diverse ancestry is expected to increase statistical power through a combination of increased phenotypic and genetic variation, larger sample sizes, as well as explanatory power with diverse patterns of linkage disequilibrium (LD)" A more careful statement is needed, e.g. "can increase" rather than "expected to increase". The "larger sample size" from combining across ancestries is only helpful if there is enough commonality in the genetic architecture. Also the "explanatory power" is better described as "improved fine mapping" which again is not guaranteed but can occur. In general including more diverse ancestries can be said to reduce power and this was why, until recently, genetic homogeneity was often sought in study designs. I agree with the need to include diverse ancestries, and this can benefit statistical power but the effects are subtle, and require a more careful discussion (or omit as not needed).

L432 I am surprised to see a high genetic correlation with "material deprivation". I'm no expert in genetics of social attributes, but it seems to me that although material deprivation may be predictable from genetic variables, this must be largely due to confounding of genetic and environmental effects and be poorly portable across ancestries, so I feel this result can't be meaningful at face value. Some further explanation is needed, or a caveat about interpretation? Also the genetic correlation with reproductive phenotypes is interesting but it would be useful to state the direction of phenotypic correlation: is smoking positively correlated with reproduction? Similarly for other genetic correlations.

L496 "according to a standard analysis plan" this sounds an implausible claim given the diversity of software listed. Where is the agreed analysis plan documented? In the brief description that

follows, analyses for unrelated individuals differ between binary and non-binary phenotypes, but there is no such difference reported for family studies, which doesn't seem reasonable.

L536 each ϵ_{jk} has its own variance term s^2_{jk} but how is this computed? Since it is specific to each j and k , the only variable left over which to compute a variance is the ancestry l , so is this a variance over ancestries? The explanation "the random effect term that captures any residual heterogeneity (i.e., error)" is not much use, and wrong because e_{jk} is the error term. A more precise statement is needed for this central aspect of the model.

L666 "Affinity propagation clustering" needs a brief explanation. Currently this sentence under "methods" adds nothing to the brief mention in the main text.

Minor issues:

L316 "five loci were overlapping for all five phenotypes", "overlapping" is the wrong word, in a science paper accuracy of wording is important.

L423 a difference of $>20\%$ is not "slightly lower".

L663 I suspect "and" should be "or", it doesn't make sense to exclude only if both conditions are true.

Author Rebuttals to Initial Comments:

We sincerely thank the referees for their thorough review and constructive comments. We have addressed each comment below with our response in bold. We have noted how and where changes were made in the manuscript and/or supplementary materials.

Referees' comments:

Referee #1 (Remarks to the Author):

Review of "Trans-Ancestry Genome-Wide Investigation of Tobacco and Alcohol Use in up to 3.4 Million Individuals" by Saunders et al.

The authors report results from genome-wide analyses of four smoking phenotypes (smoking initiation, age at smoking initiation, cigarettes per day, smoking cessation) and drinks per week across major clines of recent human ancestry (AFR, AMR, EAS, EUR) involving from 728,826 individuals (AgeSmk) up to 3,383,199 individuals (smoking initiation). Besides trans-ancestral analyses, ancestry specific analyses were conducted primarily with the aim of evaluating how polygenic scores (PGS) derived from weights obtained from ancestry specific GWASs predicted the phenotypes within and across ancestries.

The authors identified 2,143 genome-wide significant loci ($P < 5 \times 10^{-9}$) across all phenotypes, using a mixed effects meta-regression method that accounts for ancestry clines and other heterogeneities. This method demonstrated to have more power to detect loci than traditional fixed effect meta-analysis not accounting for ancestry. When using a fine-mapping method that accounts for ancestry 28% of identified loci included less than 5 credible variants. Fine-mapping was improved when including ancestry information e.g., the number of causal variants decreased on average with 20% within loci.

Overall, a small proportion of conditionally independent variants (4%) demonstrated large differences across ancestries in effect sizes. PGS analyses demonstrated EUR-based PGS in EUR ancestries to perform better than ancestry matched PGS in non-EUR ancestries.

The subject of the study i.e., cross-ancestry genetic analyses and how to improve and generate further information about the genetic architecture of complex disorders in non-EUR ancestries, is very timely and highly relevant both from a scientific and clinical point of view. The study is very well performed, carefully executed with use of the newest methods, and dubious results are addressed thoughtfully in sensitivity/follow-up analyses (e.g., the unusual genetic correlation pattern of DrnkWk driven by the 23andMe cohort). The manuscript is well written, and the results are interesting, especially I find it interesting to see the large improvement in fine-mapping, when including ancestry information.

Several of the methods applied are new to me, but as far as I could evaluate the statistic analyses are well-performed and I have no major comments or concerns regarding those. I have a few other

comments which you can find below:

1) Regarding the phenotype “Drinks per week”, it has been shown that the genetic correlation between alcohol consumption and alcohol use disorder is around 0.5 (in EUR individuals) which suggest that alcohol use disorder does not represent the extreme end of a continuous distribution of alcohol consumption when considering the underlying genetic architecture (at least in Europeans). I assume some cohorts have a high representation of individuals with alcohol use disorder. Have the authors considered that this might introduce heterogeneity in the Drinks per week phenotype?

Thank you for raising an important point. To our knowledge, recently published work on the genetic correlations between problematic alcohol use and drinks per week suggest higher genetic correlations of 0.7-0.8^{1,2}, although still far from 1.0 as the reviewer notes. In the current manuscript, using our EUR-stratified GWAS results, the genetic correlations for drinks per week were largest with questionnaire items related to problematic use in the UK Biobank (i.e., “Others concerned about, or recommend reduction of, own alcohol consumption”, “Frequency of feeling guilt or remorse after drinking alcohol in last year”, “Frequency of memory loss due to drinking alcohol in last year”), ranging from $r_g = 0.57$ to 0.70 . In comparison, the genetic correlation with “alcohol intake frequency” was $r_g = 0.74$. This is further complicated by the fact that genetic correlations between DrnkWk in one participating study and DrnkWk in another participating study was on average 0.72 in the present manuscript (Supplementary Table 7), consistent with prior work³, suggesting that even r_g between the same phenotype in different studies, while moderate to large, is also not equivalent to 1.0. As a reference, similar patterns were found in GWAS of educational attainment⁴, with an average cross-cohort genetic correlation of 0.72.

At the same time, the reviewer point stands that the observed genetic correlations suggest that while our DrnkWk phenotype and AUD are not redundant, they are moderately to strongly related measures. We have revised the manuscript to reflect this on page 11 by changing wording to “...suggesting that DrnkWk measures alcohol use, use problems, and alcohol use disorder, although it clearly does not capture all relevant aspects of use problems and disorder. This is consistent with prior findings of strong but imperfect genetic correlations (e.g., $r_g = 0.8$) between alcohol consumption and alcohol use disorder from large-scale GWAS^{1,2}.”

Finally, to the reviewer’s other point, while there are likely many people in our included cohorts that have an alcohol use disorder, only one cohort (N=9,947) was enriched for AUD. It is possible that this introduces some heterogeneity.

¹Zhou, H., Sealock, J. M., Sanchez-Roige, S., Clarke, T. K., Levey, D. F., Cheng, Z., ... & Gelernter, J. (2020). Genome-wide meta-analysis of problematic alcohol use in 435,563 individuals yields insights into biology and relationships with other traits. *Nature neuroscience*, 23(7), 809-818.

²Mallard, T. T., Savage, J. E., Johnson, E. C., Huang, Y., Edwards, A. C., Hottenga, J. J., ... & Sanchez-Roige, S. (2022). Item-level genome-wide association study of the alcohol use disorders identification test in three population-based cohorts. *American Journal of Psychiatry*, 179(1), 58-70.

³Liu, M., Jiang, Y., Wedow, R. *et al.* Association studies of up to 1.2 million individuals yield new insights into the genetic etiology of tobacco and alcohol use. *Nat Genet* 51, 237–244 (2019).

⁴Lee, J.J., Wedow, R., Okbay, A. *et al.* Gene discovery and polygenic prediction from a genome-wide association study of educational attainment in 1.1 million individuals. *Nat Genet* 50, 1112–1121 (2018).

2) When defining genes of interest, the authors apply a filtering step where only loci with better fine-mapping results in the trans-ancestry analysis compared to the EUR-stratified GWAS results were retained. I do not understand why this was required; I assume the aim would be to extract genes that are likely to affect the phenotypes in order to get a better understanding of the underlying biological factors. So why exclude loci which were nicely fine-mapped already (and did not improve further in the trans-ancestry analysis)? I think it would make most sense to focus on all loci with a set of credible variants less than five from the trans-ancestral GWAS MA. Or at least include the 192 loci where the credible set contained only one variant, no matter improvement in fine-mapping in the trans-ancestry GWAS. Additionally, the authors highlight that there was no evidence of effect moderation by ancestry for most conditionally independent variants, so I assume biological knowledge generated by focusing on all loci with few credible variants in the trans-ancestry GWAS would be relevant for all ancestries.

We agree with the reviewer and have revised the relevant sections to clarify our strategy. Our aim in subsetting variants was intended merely to provide a narrative illustration of some of the gains made specifically through the inclusion of diverse ancestries, one component of the present work. There are 1,364 variants in the 597 credible intervals containing less than five variants, which poses some difficulty for a narrative description of individual loci. To maintain a balance between quantitative summaries across all loci versus descriptions of a few loci of particular interest, we decided to include in the narrative portion of the summary only loci that were better fine-mapped in the trans-ancestry results compared to EUR-stratified results. In this way, we are highlighting loci that are well fine-mapped and for which inclusion of diverse ancestries or increased sample sizes further increases the identification of potentially causal variants. This was an editorial choice on our part, which we thought may be justifiable in order to provide a description of a few variants and genes of particular interest, versus solely focusing on quantitative summaries.

We now complement our existing narrative description with additional information for all variants from the 597 trans-ancestry loci with fewer than five variants in the credible intervals. This includes 1,364 variants across all five phenotypes, 77.6% of which show no evidence for effect size heterogeneity by ancestry. Of these variants, 118 are mapped to genes that were found in the

trans-ancestry TWAS results with 52 genes having a TWAS *P*-value below the Bonferroni threshold (corrected for 22,121 genes in 49 tissues). We have modified the text on page 9 to note these changes and have added additional columns to Supplementary Table 3 that note which variants from the well fine-mapped credible intervals are mapped to the 52 unique genes that were significant in the TWAS results.

3) Regarding my point above, the authors intersect their GWAS results with TWAS findings when identifying genes of special interest. Even though a huge amount of work has been performed in order to map out regulatory variants, the best predictor for the causal gene is still distance from the index variant to the gene. Therefore, I would recommend the authors to include genes, in their list of high priority genes, if the signal (determined by credible variants) is located within a gene, even though the TWAS results do not highlight the gene.

Intersecting the GWAS and TWAS results was done for illustrative purposes, as indicated above. However, we agree with the reviewer comment and attempt to address this concern by a) making it clear that some genes were chosen for illustration and b) including more information in Supplementary Table 3 (detailed in response to the next comment below) for all high priority genes whether or not they were significantly associated in the TWAS results.

4) Please list the high priority genes in a supplementary table together with information about the credible variants and their annotations (e.g., coding/non-coding, eQTL information etc.).

We have now provided information for all variants in the 597 loci that were fine-mapped to less than five variants in their 90% credible intervals in Supplementary Table 3. Changes to this table include additional information on which of these variants are mapped to the high priority genes along with the per variant posterior inclusion probabilities, variant annotations, TWAS cross-tissue *P*-values, and a column that flags whether each variant is an eQTL for the 52 high priority genes (defined in the response to comment #2 above).

5) I do acknowledge that the focus of the study is on how trans-ancestry genetic information can improve locus discovery and PRS analyses, rather than focusing on the biological factors underlying the phenotypes. However, the authors manage to pinpoint 597 loci with less than 5 credible variants, and 192 loci where the credible set contained only one variant. I think this is a very successful fine-mapping which should give less noise in high-priority genes mapped by credible variants and opens for the opportunity to get closer to the biology of the phenotypes. I miss further analyses that combines high priority genes with functional genomics data. Are these genes enriched in specific pathways or expressed in particular cell types that can inform us more about the underlying biological causes driving the phenotypes?

Great suggestion. We agree that enrichment analyses for the high priority genes could further our understanding of the underlying biology of substance use. We have conducted additional functional enrichment analyses in high priority genes identified in fine-mapped credible intervals. Further detail on this has been added to page 9 of the main text and on page 17 of the Supplementary Note. Briefly, we followed prior work¹ and took genes mapped from variants in credible intervals fine-mapped to less than five variants (as high priority genes) as well as genes mapped from variants with posterior inclusion probabilities (PIP) less than 0.01 (to serve as control genes), and estimated the relative risk of these genes being in several annotation categories related to tissue expression, cell type, and specific pathways. That is, a relative risk = (proportion of high priority genes that are in the annotation) / (proportion of control genes that are in the annotation). This characterizes the relative enrichment of our fine-mapped loci versus the control loci.

Full results are in Supplementary Table 11 and Supplementary Figure 7. We found that, across all five phenotypes, genes mapped from variants in the well fine-mapped credible intervals were significantly more likely to be expressed in brain tissues compared to genes mapped from variants with PIP < 0.01. Within the brain, the patterns of cell type associations closely matched findings for other psychiatric diseases and cognitive traits like schizophrenia and intelligence². Lastly, we found that genes from the small credible intervals were found more often in pathways related to neuron development, nervous system development, and synapse organization and signaling compared to genes from variants with PIP < 0.01. We comment further in the revised manuscript on the interpretation of these findings and their potential to help us understand biological etiology of substance use in the main text (page 8) and supplementary text (page 17).

¹Kanai, M., Ulirsch, J. C., Karjalainen, J., Kurki, M., Karczewski, K. J., Fauman, E., Wang, Q. S., Jacobs, H., Aguet, F., Ardlie, K. G., Kerimov, N., Alasoo, K., Benner, C., Ishigaki, K., Sakaue, S., Reilly, S., Project, T. B. J., FinnGen, Kamatani, Y., ... Finucane, H. K. (2021). *Insights from complex trait fine-mapping across diverse populations* (p. 2021.09.03.21262975). medRxiv. <https://doi.org/10.1101/2021.09.03.21262975>

²Bryois, J., Skene, N. G., Hansen, T. F., Kogelman, L., Watson, H. J., Liu, Z., Eating Disorders Working Group of the Psychiatric Genomics Consortium, International Headache Genetics Consortium, 23andMe Research Team, Brueggeman, L., Breen, G., Bulik, C. M., Arenas, E., Hjerling-Leffler, J., & Sullivan, P. F. (2020). Genetic identification of cell types underlying brain complex traits yields insights into the etiology of Parkinson's disease. *Nature genetics*, 52(5), 482–493. <https://doi.org/10.1038/s41588-020-0610-9>

Minor comments

6) In the analyses evaluating heterogeneity in effect sizes across ancestries the authors evaluate conditionally independent variants. I assume they evaluated genome-wide significant variants, but I don't think it is clear from the text, or maybe I missed it somewhere?

This is correct. We report effect size heterogeneity across ancestry for each conditionally independent variant, all of which were genome-wide significant ($P < 5 \times 10^{-9}$). We have clarified this on page 8 and in the methods section on page 14. Thank you for bringing it to our attention.

7) In the TWAS the authors correct for 22,121 genes, I think it would be appropriate to take the number of tested tissues into consideration when correcting for multiple testing.

A more conservative threshold is certainly defensible here given the number of genes, tissues, and phenotypes. We have changed the P -value threshold to $P < 4.61 \times 10^{-8}$ for the TWAS analysis which corresponds to a Bonferroni correction for 22,121 genes in 49 tissues. Changes were made to the main text on page 9, the supplementary text on page 17, and in supplementary table 5 to clarify that the P -value threshold corrected for the number of genes and number of tissues.

8) Did the authors normalize their polygenic scores?

Yes, all scores were scaled to have a mean of zero and standard deviation of one to aid in interpretation. A sentence on page 16 in the methods was added to note this.

9) What was the reason to include “age2” as a covariate in the PGS analyses?

A quadratic effect of age was included as a covariate in the polygenic risk score models based on prior knowledge of the developmental trajectories of smoking and alcohol use^{1,2}. For measures of tobacco and alcohol use there are nonlinear relationships with age, with a steep increase in frequency and quantity of use starting in early adolescence followed by a peak in early adulthood and either stabilization or a slight decrease in use throughout later adulthood. Because of this, a quadratic effect of age was expected to result in improved model fit and increased power to detect a polygenic score effect. After fitting the polygenic scores models, we found a significant quadratic effect of age in only one model though we decided to keep this covariate term in each model to avoid, to the extent possible, researcher degrees of freedom.

¹Kendler, K. S., Schmitt, E., Aggen, S. H., & Prescott, C. A. (2008). Genetic and environmental influences on alcohol, caffeine, cannabis, and nicotine use from early adolescence to middle adulthood. *Archives of general psychiatry*, 65(6), 674–682.
<https://doi.org/10.1001/archpsyc.65.6.674>

²Vrieze SI, Hicks BM, Iacono WG, McGue M. Decline in genetic influence on the co-occurrence of alcohol, marijuana, and nicotine dependence symptoms from age 14 to 29. *Am J Psychiatry*. 2012;169(10):1073-1081. doi:10.1176/appi.ajp.2012.11081268

10) In supplementary table 6, could the authors include a column indicating if the heritability estimates are significantly different from zero.

Thanks for the suggestion. Using a normal approximation (i.e., $h^2 \pm 1.96 * SE$), only the SNP heritability estimate for AgeSmk in AFR ancestry LDSC analysis was not significantly different from zero. We have added a sentence to the table note highlighting caution in interpretation of the AFR ancestry AgeSmk results, and to a lesser extent CigDay results, due to imprecision caused by relatively small sample sizes.

11) Supplementary Figure 4, B1, the y-axis is named "PC2*(-1)" should that be "PC1*(-1)"?

Thank you for pointing this out! The axis label has now been fixed.

12) Supplementary table 6, there is an unusual high ratio in the AFR, AgeInit analysis (ratio = 0.609). This indicates that a very high proportion of the inflation in the test statistics is caused by other factors than polygenic heritability. I think this is unusual high, could the authors comment on this?

The African ancestry-stratified analysis for AgeSmk unfortunately has the smallest sample size of any phenotype-ancestry combination (N=17,518 for the LDSC analyses). This results in larger standard error estimates for SNP heritabilities, intercepts, and attenuation ratios than for other phenotypes. The attenuation ratio is also particularly important for genetic association results based on very large samples (e.g., >500K), where bias is induced in the LDSC intercept¹. Sample sizes are so small for AgeSmk and CigDay in AFR ancestries we suggest the reader pay closer attention to the genomic control and intercept directly and treat the attenuation ratio as far less relevant. Taken together, the AgeSmk and CigDay LDSC analyses in AFR-stratified samples may be somewhat underpowered and likely not the most relevant statistic for the reader to consider. We have added a sentence to the Supplementary Table 6 note that directs readers to Supplementary Text page 14 where we comment on this.

¹Loh, P.-R., Kichaev, G., Gazal, S., Schoech, A. P. & Price, A. L. Mixed-model association for biobank-scale datasets. *Nature Genetics* 50, 906–908 (2018).

Referee #2 (Remarks to the Author):

Overview: Saunders, Wang, Chen, Jang, Liu, Wang, et al. present their work investigating the genetic

contribution to tobacco and alcohol use phenotypes in a large-scale multi-ancestry analysis. Overall they report numerous genomic risk regions associated with these phenotypes, the majority of which are consistently identified across ancestry groups. In line with prior works, multi-ancestry fine-mapping considerably reduces credible SNP sets, which are used to illustrate several genes with putative effects on smoking phenotypes identified through TWAS. Their work contains numerous statistical advancements, which will have broad applicability to the field of multi-ancestry statistical genetics. The manuscript is well written, and the results presented in a clear, understandable manner. The authors have performed detailed validation when appropriate and I commend them for their overall efforts in combining numerous genetic and genomic datasets. I do have a few minor comments related to some of the claims.

Major Comments:

1. "Some of these variants unique to trans-ancestry results are located in genes related to neurotransmission (e.g., *NRXN1*) including glutamatergic (*GRIN2A*) neurotransmitter systems, of relevance to neurocircuitry in addiction^{11,12}."

The authors report that their meta-regression approach improves power to detect risk regions, and provide support for this fact with some risk variants falling near genes related to neurotransmission. What is unclear to me is if this is expected by chance or not, given the large increase in the number of associations found. It would be helpful if the authors could perform some permutation test or enrichment test to provide stronger statistical support for these findings.

In reviewing this paragraph in light of a comment from Reviewer 4, we concluded that a slight rewrite of this paragraph may be helpful. As noted by the reviewer, we described here power gains due to improved modeling of genetic ancestry, more than we described gains due to diverse ancestry versus, say, a meta-sample composed entire of EUR ancestries. We have modified the paragraph to more clearly describe that we found 721 independent variants in the full trans-ancestry model that were not identified in the simpler fixed-effects model. We later (second full paragraph on page 8) also describe the gains in fine-mapping resolution due to inclusion of diverse ancestries, finding that almost half of the decrease in credible interval size between trans-ancestry and EUR-stratified fine-mapping results is attributable to inclusion of diverse ancestries. The other half is likely due to increased sample sizes. We are grateful you brought this issue to our attention.

2. "There was no evidence of effect moderation by ancestry for most conditionally independent variants, ranging from 79.2% (194 variants) in CigDay, to 89.9% (187 variants) in SmkCes. Another 10.3% to 16% showed modest evidence for allelic heterogeneity. Finally, roughly 4% of all conditionally independent variants, reflecting 137 variants from 86 distinct loci, showed strong evidence of effect size moderated by ancestry (complete results in Supplementary Table 2)."

I was glad to see this analysis performed, which provides strong support for the overall meta-regression and meta-analysis across ancestries. While a small percentage exhibited evidence of heterogeneity, can the authors demonstrate that these are not driven by differential genotyping

quality/imputation at these regions. Furthermore, can the authors report F_{st} at these sites compared to the background? Similarly, can the authors report the ancestry-matched LD scores at these sites compared to the background?

Additional analyses were performed to compare imputation quality, LD scores, and F_{st} between variants that showed strong evidence of heterogeneity and those with no evidence of heterogeneity for each ancestry. That is, for every conditionally independent variant we computed per ancestry LD scores using all TOPMed cohorts that contributed to the meta-analysis and calculated the mean imputation quality scores for each ancestry across all contributing cohorts to the SmkInit GWAS. Additionally, we again used the TOPMed cohorts to compute F_{st} values for each ancestry pair (i.e., pairwise EUR-EAS, EUR-AFR, EUR-AMR, AFR-AMR, AFR-EAS, and AMR-EAS). We then compared the distributions of each of these (LD scores, imputation quality, pairwise F_{st}) between variants with no evidence of heterogeneity and variants with strong evidence of heterogeneity. We also compared the distributions between variants that were heterogeneous on each MDS component and those with no evidence of heterogeneity. For example, we compared the 88 variants showing allelic heterogeneity across MDS component 1 and the 3,032 variants showing no evidence of heterogeneity on any MDS component).

In total there were 70 mean comparisons with 3 P -values below the Bonferroni corrected threshold including (1) significantly lower imputation quality for variants heterogeneous on MDS component 1, the EAS cline (mean [M] = .95, standard deviation [SD] = .09), compared to variants with no evidence of heterogeneity (M = .97, SD = .04), $t(111.6) = -5.11$, $P = 1.36e-6$; (2) significantly greater pairwise F_{st} between AFR and EAS ancestries for variants heterogeneous on MDS component 3, the EUR cline (M = .12, SD = .14), compared to variants with no evidence of heterogeneity (M = .02, SD = .02), $t(9.94) = 13.95$, $P = 7.49e-8$; and (3) significantly greater pairwise F_{st} between AFR and AMR ancestries for variants heterogeneous on MDS component 4, the AMR cline (M = .07, SD = .09), compared to variants with no evidence of heterogeneity (M = .02, SD = .03), $t(9.75) = 5.71$, $P = .0002$.

The small number of mean differences and a lack of clear pattern of results suggests that the identification of heterogeneous variants was not driven to a large extent by differential imputation quality, LD scores, or F_{st} . We're grateful to the reviewer for suggesting these analyses and describe our results in two paragraphs on page 16 of the supplementary note.

Minor Comments:

1. Their cross-ancestry replication method appears to provide support for the overwhelming majority of genomic risk regions, which is consistent with intuition and previous studies, however there were a small number that failed to replicate (17 variants with $prob < 0.99$). Can the authors provide some additional information on how their model decided that these variants had less evidence of replication.

We thank the reviewer for the comment. The cross-ancestry replication method, termed RATES, relies on a mixture model to accommodate three scenarios of marginal genetic effect sizes: 1) the variant has a null effect; 2) the variant is an outlier with an inflated effect size; 3) the variant has a genuine non-zero effect size. For each variant, we assign them to one of the three scenarios and calculate the posterior probability that the variant has a genuine non-zero effect, which we call the posterior probability of replicability. Variants that have strong and consistent effects across studies (after adjusting for ancestral differences) are more likely to have genuine non-zero effects and will have a higher posterior probability of replicability.

We constructed additional plots for the 17 variants that have a posterior probability < 0.99 and included them in Supplementary Figure 8. Each plot summarizes the Z-score distribution from available studies. The color represents ancestry, and the point size represents the sample size of each cohort. When the underlying genetic effect is similar, the magnitude of the Z-score statistic should be proportional to the square root of the sample size for non-outlier SNPs. Yet, for these variants with low probability of replicability, we observe that the significance of meta-analysis results is driven by a few outlier studies. For example, in SmkCes, for the variant chr2:10619084_G/A, only one study has a disproportionately large Z-score of 34.93 and all the other studies have very small Z-score statistics.

2. The TESLA method appears to apply eQTL prediction weights to the GWAS effect-sizes adjusted for sampling error (the fitted effects from MEMO), but do not include the heterogeneity term (random effect term in MEMO). This doesn't appear to have a justification, and I was somewhat puzzled given the primary GWAS results are reported using the random-term.

The reviewer is correct that MEMO decomposes the genetic effects into a component that is invariant across ancestries (intercept), a component that varies with the ancestry (fixed effect terms with MDS ancestry components), and a component that varies independently of ancestry (random effect). Given a set of eQTL weights from a given ancestry, it is most powerful to integrate it with a GWAS dataset of the same ancestry (e.g., European ancestry in both datasets), and this kind of ancestry matching has now long been a standard approach in TWAS. When faced with more ancestrally diverse data, we developed TESLA to more accurately estimate genetic effects in samples of differing ancestries using the meta-regression model to map eQTL datasets with GWAS studies with any ancestry composition. To do this mapping, we only need the component that is invariant across ancestries (i.e., the intercept term) and the components that are specific to ancestry (i.e., fixed effect terms), but not the random effect that is by definition independent of ancestry. That is why TESLA does not include the random effect term in its model, as it cannot be mapped across ancestries. Ultimately, the decision to exclude the random effect in the present analyses may be further justified because the random effects in the MEMO results themselves were typically extremely small, and removing an unnecessary term will increase power.

Referee #3 (Remarks to the Author):

Review of “Trans-Ancestry Genome-Wide Investigation of Tobacco and Alcohol Use in up to 3.4 Million Individuals”

This is a follow-up study of the GSCAN study published in 2019 in Nature Genetics, which included 1.2M individuals. The major novelties of the current study are the impressive sample size and the inclusion of diverse ancestries. The authors conduct a GWAS in 3.3M individual and identify thousands of loci linked to tobacco and alcohol consumption. I believe this article provides important new insights into the genetics of alcohol and tobacco consumption and will be of interest to readers. The manuscript is well written and the methods are appropriate. The results are presented in a clear and concise manner. Generally, the use of statistics is appropriate and uncertainties well described except for the PRS analysis, I believe (see specific comments below). I have some suggestions for improvement.

1) it seems that no covariates (e.g., gender, SES) were included in the GWAS. Can a statement be included to clarify this, in the methods section of the main text (and potentially also in the results section). I believe this is important information for interpretation.

Contributing studies included the covariates of sex, age, age², and genetic principal components when generating summary statistics, and were instructed to include additional covariates as may be applicable to their specific study (e.g., ascertainment). The meta-analysis did not include additional covariates beyond the meta-regression ancestry terms. Further information in the generation of individual summary statistics is included in the supplementary text. We have added a statement about the covariates that were included in the methods section on page 13. Thank you for raising this, and the opportunity to ensure the models are clearly communicated in the text.

2) Do any of the MDS components potentially represent social/cultural factors? E.g., there are large cultural differences in drinking behaviors within Europe, so it may be worth investigating potential confounding factors

The MDS components could represent some social, cultural, or environmental factors as they are based on allele frequencies across ancestries. Unfortunately, we have no direct way of investigating potential confounding factors like cultural differences in drinking behaviors or tobacco use. We note, though, that the random effect term in the trans-ancestry GWAS model captures residual heterogeneity in effect sizes that is not due to the first four MDS components. In this way, the model provides some accounting for the effect of social/cultural factors, at least those independent of ancestry, when estimating variant effect sizes. Disentangling the relative contribution of ancestry and cultural factors is a thorny issue that will require substantial follow-

up research and investment, although the present manuscript arguably provides a crucial step toward understanding such issues. We now note, at the end of the discussion section on page 12, that further considerations of the effects of cultural/environmental factors will be critical to understanding the genetic architecture of substance use.

3) TWAS analyses lead to the identification of many genes. I believe it would be of interest to further explore the biological mechanisms implicated by these genes, e.g., through pathway or gene-set analyses.

Thank you for this suggestion. We agree that enrichment analyses of the TWAS results could further our understanding of the biological mechanisms implicated by identified genes and have now added pathway analyses. Details on this approach and results are now included on page 9 of the main text, pages 18-19 of the supplementary note, and a new supplementary table 12. Briefly, we conducted pathway enrichment analyses similar to MAGMA, but instead of gene-level statistics from single SNP *P*-values in GWAS, the input statistic now is based upon TESLA TWAS *P*-values from each tissue. Full details of the trans-ancestry TWAS method, termed TESLA, and the pathway enrichment analysis, termed eTESLA, are included in a forthcoming paper by Chen et al.¹.

Briefly, TWAS pathway enrichment results identified potentially important biological pathways for tobacco and alcohol use phenotypes, which were broadly enriched across multiple tissues. For example, acetylcholine-gated channel pathways were enriched in multiple brain tissues for SmkInit, CigDay, and SmkCes. Behavioral response to nicotine pathways, that similarly contain nicotinic receptor (*CHRN*) genes, were significantly enriched broadly across tissues for CigDay and SmkCes. Dopamine receptor signaling and binding pathways, of obvious relevance to neurotransmission, were significantly enriched across tissues for SmkInit, CigDay, and DrnkWk, and alcohol dehydrogenase activity pathways for DrnkWk were enriched in 26 tissues, including 7 (of 13) brain tissues. Full results for all significantly enriched pathways have now been added to Supplementary Table 12.

¹Fang Chen, Xingyan Wang, Seon-Kyeong Jang, J. Dylan Weissenkampen, Chachrit Khunsriraksakul, Lina Yang, Renan Sauteraud, ..., Dana Hancock, Bibo Jiang, Scott Vrieze, Dajiang J. Liu: "Trans-ethnic Transcriptome-wide Association Study for Smoking Addiction in 1.3 Million Individuals Yields Insights into Tobacco Use Biology and Drug Repurposing". Fourth revision under review at *Nature Genetics*.

4) Can the authors provide additional information on the putative causal variants in the CHRNA5-A3-B4 locus? Did their analysis allow for pinpointing the causal molecular mechanisms that underlie this association? This has been widely debated in the field, and I feel these data provide a unique opportunity to establish the putative causal genes in this locus (this has been difficult previously, due to high LD)

This is a great point, and we agree the present data can further our understanding of this locus, although perhaps an experimental molecular biology program of research will likely be required to establish causal variants/genes and molecular mechanisms underlying the association. However, in the revision we have expanded the discussion of this locus. Specifically, the two variants within the credible interval of the *CHRNA5-A3-B4* locus include rs2869055 and rs28438420 for Smklnit. Interestingly, these variants are not in high LD with a putative causal variant (rs16969968; $R^2 = 0.31$ for both variants) while those in the CigDay credible interval for this locus (fine-mapped to two variants) were in high LD with rs16969968 ($R^2 = 0.84$ and 0.86). It appears that the variants underlying this signal for smoking initiation seem to be somewhat distinct from those for cigarettes per day. Text has been added to page 9 to note the differences in LD between Smklnit and CigDay and the known causal variant.

5) Lines 449-450 (comparison of PRS across ancestries). Can more info on statistics be included? I would expect low statistical power to detect differences in predictive accuracy, given the relatively small sample sizes of the target samples. The results in Figure 2 confirm this, SE's are large and often overlap across ancestries. Many SE's (especially for non-European populations) include 0. I'm not sure the results as described in the main text appropriately acknowledge the limited power to identify differences.

We agree that statistical power affects confidence in interpreting these results, especially for evaluating differences in the variance explained by scores across some of the non-EUR target ancestries. We have added a note about this on page 11, in the caption for figure 2, and further detail regarding power analysis on page 22 of the supplementary text.

6) Lines 468 and further. I believe the interpretation of these findings as being in line with the omnigenic model is a stretch. The authors have not differentiated between core and peripheral genes in their analyses. For this, integration of gene expression networks would be required. I believe that the discussed should be toned down, or additional analyses would need to be performed.

We have toned down the wording on page 11 to note briefly only that others have argued that the discrepancy between lack of allelic effect size heterogeneity and the poor cross-ancestry PRS prediction may be consistent with the omnigenic model (argued most recently and prominently by Ian Mathieson¹).

¹Mathieson, I. The omnigenic model and polygenic prediction of complex traits. *The American Journal of Human Genetics* 108, 1558–1563 (2021).

7) Lines 479-481. I don't agree that exposure to tobacco and alcohol will be similar for all study participants, variation in culture and socio-economic status will be important. I do agree that variation may be even more extensive at a global level, so a slight modification of these sentences would probably be sufficient.

It's a good point. We have attempted to use more precise language in the revision. We have removed the word 'culture' as we agree that despite living within the same country, individuals may still differ substantially in cultural environments. We have added the term 'availability' in its place (page 12) as our intent with the original sentence was to note that despite differences in genetic ancestry most individuals in the current study live in the U.S. (or Europe) where tobacco and alcohol are legal and largely available in the same ways for all or at least a great majority of individuals.

8) Can the authors comment on the suitability of their GWAS method in the context of admixed samples. Would it be worth comparing to other recent approaches, e.g., tractor? This is not my area of expertise, but a few sentences on how these methods compare may be helpful

Thank you for this question. It is clearly an interesting issue and one that is under active development. Recent association methods for admixed samples using local ancestry, like Tractor, are not so much competing methods, but complementary to our trans-ancestry GWAS approach. Tractor, for example, can only be applied to individual-level data. It may be useful for an individual cohort of admixed ancestry individuals to generate summary statistics using Tractor for inclusion in a meta-analysis but given that we only have study level information we are not able to compare this to our meta-regression approach. Unfortunately, methods like Tractor were not available when our participating cohorts were conducting their GWAS to contribute to our meta-analysis. We have added text discussing the above on page 10 of the supplementary text noting how the current approach differs, but may be complementary to, other GWAS approaches in the context of admixed populations.

We also note in the supplement that GWAS methods in admixed samples are an area still under active development, as may be evidenced by recent competing commentaries to the original Tractor publication^{1,2,3}, which apparently generated some mild controversy within the field. In fact, a rebuttal² to the Tractor showed that a linear model that adjusts for global ancestry with principal components is consistently the most powerful method. Arguably, it may remain to be seen whether such methods are truly more powerful, from a statistical or explanatory standpoint. We expect to see further interesting developments on this point in the next few years.

¹Atkinson, E. G., Maihofer, A. X., Kanai, M., Martin, A. R., Karczewski, K. J., Santoro, M. L., ... & Neale, B. M. (2021). Tractor uses local ancestry to enable the inclusion of admixed individuals in GWAS and to boost power. *Nature genetics*, 53(2), 195-204.

²Hou, K., Bhattacharya, A., Mester, R., Burch, K. S., & Pasaniuc, B. (2021). On powerful GWAS in admixed populations. *Nature genetics*, 53(12), 1631-1633.

³Atkinson, E. G., Bloemendal, A., Maihofer, A. X., Nievergelt, C. M., Daly, M. J., & Neale, B. M. (2021). Reply to: On powerful GWAS in admixed populations. *Nature genetics*, 53(12), 1634-1635.

9) Figure 2: I find panel A difficult to read, colors can't be distinguished in the AFR target sample. Also, does the SE of the colored bars represent the SE of all ancestries combined? The error bars seem to overlap across the different ancestry target samples and tend to be very large, often including zero (except for the European target sample). Is the power sufficient to interpret these differences? (related to comment 5)

We have made changes to the wording of the figure note to better describe that the error bars represent the 95% confidence intervals of the variance explained by the PRSs of all ancestries combined (it is too visually crowded to provide SEs for each ancestry individually, although those SEs are provided in Supplementary Table 9), and that some colors cannot be easily distinguished because there is limited incremental variance explained by a particular PRS. We have also added a note regarding the possible issues of power that are further explained in the Supplementary Note page 22.

Minor:

- Lines 367-372, please explain that information is combined across tissues as explained in detail in the methods section

We thank the reviewer for this comment. We used a Cauchy combination test to combine the *P*-values from multiple tissues and generate a combined *P*-value for each gene. We choose the Cauchy combination test as it is relatively powerful yet simple to implement. This method is also well-suited for combining *P*-values under a correlational structure and is computationally tractable with large quantities of data. We changed the original text (page 16) to: "We used a Cauchy combination test¹ to combine *P*-values across all available tissues for each gene". We also changed the corresponding supplementary text (page 18) to: "Based on multi-tissue TWAS results using the Cauchy combination test, we found 41, 1,474, 300, 251, and 667 unique genes associated with AgeSmk, SmkInit, CigDay, SmkCes, and DrnkWk, respectively."

¹Liu, Y., & Xie, J. (2020). Cauchy combination test: a powerful test with analytic p-value calculation under arbitrary dependency structures. *Journal of the American Statistical Association*, 115(529), 393-402.

- Supplementary tables: Please provide more extensive explanation, define abbreviations and define information in columns.

This is a great point. We have included more information and explanations of column abbreviations in each supplementary table. We have also added information in the table notes about which sections of the Supplementary Text contain more detail on the methods and results.

- Line 414: I suggest to add “genetic correlation” to the title of this section

We have changed the section title as suggested from ‘Genetic architecture and polygenic scoring’ to ‘Genetic architecture, genetic correlation, and polygenic scoring’.

Referee #4 (Remarks to the Author):

This is potentially an important manuscript, and could be appropriate for publication in Nature. I am broadly happy with the analysis methods as I understand them, and I applaud the emphasis on treating ancestry as continuous rather than discrete (though there still remains a substantial amount of analysis in ancestry groups, which is reasonable).

However I am concerned about multiple confusing statements describing methods that indicate lack of understanding and/or care. None of these is individually a major cause for concern but collectively they raise a concern about the authors' grasp of the methods employed, which are often not well explained so that there remain important aspects of the analysis methods that I do not understand.

Some examples of wording that causes concern:

L310 "posterior probability of replication $> .99$ " this doesn't make sense to me, replication probability depends on the replication study and it makes no sense to try to report it. I suspect that what they are reporting is a posterior probability of a non-zero effect that is large enough that a well-powered replication study will detect it, but this requires at least a brief explanation in the main text and more details elsewhere.

Thank you for this comment. The replication method, termed RATES, extends MAMBA¹ and calculates a posterior probability that a variant has a genuine non-zero effect (what we had called the posterior probability of replicability) and can be replicated in a dataset with sufficiently large sample size. As in MAMBA, RATES can also improve the effect size estimates by borrowing strength across variants. With estimated effect sizes, we can estimate the sample size needed to

replicate a certain association signal with a given power threshold or estimate the power for a given sample size.

¹McGuire, D., Jiang, Y., Liu, M. et al. Model-based assessment of replicability for genome-wide association meta-analysis. *Nat Commun* 12, 1964 (2021).

L311 This is the first of many instances of the phrase "conditionally independent variants" which doesn't make sense to me. Conditional independence involves three variables: A and B may be unconditionally dependent but independent conditional on C. I suspect that the authors just mean "independent", not "conditionally independent", as independence is often checked by a conditioning analysis: if A and B are independent causal variants, then the effect size of A is not affected by conditioning on B and vice versa.

Yes, "independent" is likely more straightforward and we have adopted that term in the revision.

L316 "high expected replicability" similar to L310, I don't understand this. Once again I think the authors may mean a high posterior probability of an effect size above some threshold.

Thank you for catching this. We did indeed mean a high posterior probability of replicability. This wording has been appropriately changed (page 8).

L325 "2,186 variants that were identified only by ... illustrating an increase in power." How many variants were only identified by the alternative analysis? No mention seems to imply zero but I am sceptical about this, and if non-zero then not reporting it is an omission that could be seriously misleading. Also increased hits do not prove an increase in power, better to be restrained and not claim this - it isn't needed.

We have rewritten this paragraph for clarity and conducted additional analyses to better describe the effect of diverse ancestries on fine-mapping resolution as well as increases in power to detect associations by properly accounting for heterogeneity of allelic effects correlated with ancestry, as our current approach does. More detail is given in our response to the 'L320' comment below.

L331 "ancestral variation in LD" another confusing phrase, possibly what is meant is variation in LD across ancestry groups? We can't measure LD in ancestral pops.

We have changed the wording (page 8) to be clear that we did indeed mean variation in LD across ancestries.

L332 "90% credible intervals of the posterior probability of inclusion" Another worrying misuse of terms suggesting poor understanding: a probability interval for a probability doesn't make much sense. I guess from context that what is meant is a credible interval for the location of a causal variant.

Thank you for catching this. We have removed 'of the posterior probability of inclusion'.

L334 "a median of 9–19 variants" a median is a single number. As you present a range, I guess you are reporting the median for several datasets, presumably one for each ancestry group but this needs clarity.

We have added "across phenotypes" on page 8 to denote that the range of medians refers to the medians for each of our five phenotypes.

L346 this is the first of many uses of the word "moderated" without clarification of what that means, which can be gleaned from methods but it isn't straightforward. In effect I think "moderated by ancestry" means that the effect size differs significantly across ancestry groups.

We have changed the first sentence that uses "moderated" to further detail what we mean in this context. Wording has been changed to "...whether a variant effect size differed (i.e., was moderated) by ancestry along four ancestry dimensions estimated from multi-dimensional scaling...". Later uses of the term are all followed by 'by ancestry' to maintain this improved clarity.

L363 "take two people who both carry one copy of the protective T allele for this variant but are separated by one standard deviation on MDS component 1 ... **despite the same genotype.**"
Confusing wording because the large difference on the MDS component means that they have many differences in genotype. Omit the last 4 words and the sentence is fine.

Changed to "despite the same rs1229984 genotype in *ADH1B*" on page 9. The full sentence now reads, "The person with a lower value on that MDS cline would be expected to drink 0.3 fewer drinks than the person with a higher MDS value, despite the same rs1229984 genotype in *ADH1B*."

L435 "highlighting their relevance to adverse disease outcomes" seems to misunderstand genetic correlations. The high correlation says that the smoking variables and the disease outcomes have to a large extent the same genetic causes, which in fact diminishes their "relevance", suggesting that smoking variables are not directly causal for disease but in fact both are caused by (largely) the same genetic mechanisms.

Good point. We meant “relevance” in that smoking and adverse disease outcomes are linked in the ways suggested by the reviewer (sharing genetic causes) without precluding that they can be linked in other ways. Changed “relevance” to “genetic link” (page 11).

L444 ".. over and above scores already entered into the model." I do not understand this, are we talking about polygenic scores (if so how do they differ from the new ones) or do you mean non-polygenic regression predictors (in which case best not call them scores). I also do not understand "EUR validation ancestries" here (L446), eventually it is explained in methods but not clear here.

Wording has been added on page 10 to clarify that "...over and above scores already entered into the model" refers to a set of iterative models where the first includes, for example, the AMR-based score and the following add the AFR-, EAS-, and EUR-based scores one at a time. At each step we evaluated the increased (incremental) predictive accuracy of the score just added. We have removed the word ‘validation’ throughout this section and replaced it with ‘target sample’ to note that this means the ancestry group of the independent target sample (Add Health cohort here).

L470 "core disease-associated pathways (e.g., those that are genome-wide significant)" this seems to misunderstand the concept of core pathways, statistical significance is not relevant to understanding the concept.

We agree and have removed the reference to ‘genome-wide significant’ on page 11. We simply meant that the variants affecting core pathways were genome-wide significant, not that the pathway itself was.

L546 More poor wording "inflation of test statistics is expected with high polygenicity" if there are many large test statistics due to polygenicity, this is not "inflation" which refers to spuriously large values e.g. due to confounding. Better just to say that GC assumes sparsity which is generally inappropriate but may be OK for low-MAF variants

We agree again and have changed wording on page 13 from “inflation of test statistics” to “elevation of genomic control values” and have noted that GC assumes sparsity. Thank you for the suggestion.

L928 (Fig 2) The caption for Panel (c) is incomprehensible to me. I don't understand the model description and the "for example" doesn't seem to relate to the model description that precedes it

We have rewritten a major portion of this figure caption to clarify what is illustrated in Panel (c).

Other issues:

L320 "Diverse ancestry is expected to increase statistical power through a combination of increased phenotypic and genetic variation, larger sample sizes, as well as explanatory power with diverse patterns of linkage disequilibrium (LD)" A more careful statement is needed, e.g. "can increase" rather than "expected to increase". The "larger sample size" from combining across ancestries is only helpful if there is enough commonality in the genetic architecture. Also the "explanatory power" is better described as "improved fine mapping" which again is not guaranteed but can occur. In general including more diverse ancestries can be said to reduce power and this was why, until recently, genetic homogeneity was often sought in study designs. I agree with the need to include diverse ancestries, and this can benefit statistical power but the effects are subtle, and require a more careful discussion (or omit as not needed).

In reviewing this paragraph, we found that the language and structure could have been clearer and are grateful that you brought this issue to our attention. Instead of describing here the gains due to genetic diversity, we had intended to describe the increase in power to detect associated variants between our full trans-ancestry meta-regression model and a simpler trans-ancestry fixed effects model (i.e., gains from different modeling, not different data). We have significantly changed the wording of this paragraph (first full paragraph on page 8) to clarify what was originally meant. In short, we find 721 independent variants in the full trans-ancestry model that were not identified in the simpler fixed-effects model. We later (second full paragraph on page 8) describe the gains in fine-mapping resolution due to inclusion of diverse ancestries, finding that almost half of the decrease in credible interval size between trans-ancestry and EUR-stratified fine-mapping results is attributable to inclusion of diverse ancestries. The other half is likely due to increased sample sizes. We are grateful you brought this issue to our attention.

L432 I am surprised to see a high genetic correlation with "material deprivation". I'm no expert in genetics of social attributes, but it seems to me that although material deprivation may be predictable from genetic variables, this must be largely due to confounding of genetic and environmental effects and be poorly portable across ancestries, so I feel this result can't be meaningful at face value. Some further explanation is needed, or a caveat about interpretation? Also the genetic correlation with reproductive phenotypes is interesting but it would be useful to state the direction of phenotypic correlation: is smoking positively correlated with reproduction? Similarly for other genetic correlations.

This is a good point. We have added words to denote the direction of genetic correlations throughout this section of the manuscript. We have also clarified that the measure of "material deprivation" used was based on neighborhood characteristics (the Townsend deprivation index in the UK Biobank) which, as you note, may reflect gene-environment confounding. A sentence has been added to page 10 highlighting the difficulties of interpreting genetic correlations broadly. We

do not have a way to assess the portability of genetic correlations across ancestry but we do agree that they can certainly be affected by genetic confounding and should not be interpreted as necessarily representing a causal relationship.

L496 "according to a standard analysis plan" this sounds an implausible claim given the diversity of software listed. Where is the agreed analysis plan documented? In the brief description that follows, analyses for unrelated individuals differ between binary and non-binary phenotypes, but there is no such difference reported for family studies, which doesn't seem reasonable.

While we provided a general description of the GWAS analysis plan in the methods section, we reserved greater detail for the supplementary text (pages 4-6). We have removed the term 'standard' to avoid misinterpretation. Although our analysis plan for the generation of summary statistics by participating cohorts has been adapted for use by other genetic association consortia, it is not standard in the sense that a defined community of scientists has formally vetted and approved it as a standard. Regarding the differences in analytical choices among samples of family or unrelated individuals, and the use of linear or logistic models, large multi-site studies like the present one strike balances between methodological and practical concerns. Many participating studies were quite large, necessitating the use of computationally efficient approaches (in this case mixed-models) that could simultaneously account for complex family structures through use of a kinship matrix as a random effect. This approach works well for genetic association studies of binary phenotypes with high sample prevalence^{1,2} (as is true here), has been used successfully previously^{3,4}, and allows ready inclusion of all individuals within a family sample. In the meta-analysis, the per-study statistics were normalized prior to meta-analysis to help ensure comparability of effect across studies.

¹Cook, J., Mahajan, A. & Morris, A. Guidance for the utility of linear models in meta-analysis of genetic association studies of binary phenotypes. *Eur J Hum Genet* 25, 240–245 (2017).

²Kang HM, Sul JH, Service SK *et al*: Variance component model to account for sample structure in genome-wide association studies. *Nat Genet* 2010; 42: 348–354.

³Xue, A., Wu, Y., Zhu, Z. *et al*. Genome-wide association analyses identify 143 risk variants and putative regulatory mechanisms for type 2 diabetes. *Nat Commun* 9, 2941 (2018).

⁴Fuchsberger, C., Flannick, J., Teslovich, T. *et al*. The genetic architecture of type 2 diabetes. *Nature* 536, 41–47 (2016).

L536 each ϵ_{jk} has its own variance term s^2_{jk} but how is this computed? Since it is specific to each j and k , the only variable left over which to compute a variance is the ancestry l , so is this a variance over ancestries? The explanation "the random effect term that captures any residual heterogeneity (i.e., error)" is not much use, and wrong because ϵ_{jk} is the error term. A more precise statement is needed for this central aspect of the model.

We thank the reviewer for the comment. The error term follows $\epsilon_{jk} \sim N(0, s_{jk}^2)$, where s_{jk}^2 is the variance of the genetic effect estimate b_{jk} . Both b_{jk} and s_{jk}^2 are available as part of the contributed summary statistics from each study in the meta-analysis. e_{jk} is a random effect in the meta-regression model. It captures the genetic effect heterogeneities across cohorts that are not explained by the MDS components. We have now clarified on page 14 of the main text and page 9 of the supplementary text that e_{jk} denotes the random effect, ϵ_{jk} denotes the error term, and s_{jk}^2 is the variance of b_{jk} taken from the per study summary statistics.

L666 "Affinity propagation clustering" needs a brief explanation. Currently this sentence under "methods" adds nothing to the brief mention in the main text.

Words have been added on page 15: "a message passing algorithm based on exemplars that identifies their corresponding set of clusters".

Minor issues:

L316 "five loci were overlapping for all five phenotypes", "overlapping" is the wrong word, in a science paper accuracy of wording is important.

We define a locus as a genomic region with a start and end position based on LD estimated from an ancestry matched TOPMed reference panel (described on page 15). By 'overlapping' we mean that the end base pair position of one locus stretches beyond the start base pair position of another locus. So by "five loci were overlapping for all five phenotypes", we mean that there are five loci that are common to each of the five phenotypes. We have changed the wording in the main text to "...five loci were associated with all five phenotypes" to avoid confusion.

L423 a difference of >20% is not "slightly lower".

We have removed the modifier "slightly".

L663 I suspect "and" should be "or", it doesn't make sense to exclude only if both conditions are true.

Thank you for catching this! It has now been corrected by changing 'and' to 'or'.

Reviewer Reports on the First Revision:

Referees' comments:

Referee #1 (Remarks to the Author):

First, I would like to congratulate the authors with their very interesting and comprehensive work. There are many new and exciting findings which include loci discovery, effect size modulation by ancestry, improved fine-mapping when including ancestry information and careful evaluation of PRS prediction across ancestries. In my opinion, the study merits publication in Nature.

The authors have cleared most of my points raised. I only have a few minor comments:

1) I have an additional comment regarding the DrnkWk phenotype. In the current study the authors find a non-significant small positive genetic correlation of DrnkWk with college degree ($r_g = 0.04$, $P = 0.03$) and job codes related to higher education (reflecting other studies finding alcohol consumption to have a positive genetic correlation with educational attainment (Kranzler et al. 2019)). Alcohol use disorder is known to demonstrate, a significant negative genetic correlation with educational attainment. So, I would be cautious stating that DrnkWk "measures" alcohol use disorder because there is a part of the genetic component underlying DrnkWk that differs from that underlying AUD (which the authors also highlight). I would therefore suggest a more moderate wording simply stating that there exist a genetic overlap, e.g., : "DrnkWk was most strongly correlated with problematic drinking behaviors ($r_g = 0.52-0.70$), suggesting an overlap in the genetic architecture of DrnkWk and measures of alcohol use problems".

2) Please also state the P-value threshold for claiming significant genetic correlation, I could not find that in the text.

3) I am very happy to see that the authors have included enrichment analyses of high priority genes. Thank you very much! I have a few questions related to this. On page 8 the authors write: "To characterize genes prioritized from fine-mapping, we conducted a series of functional....". It is not clear how genes are linked to credible variants – do the authors require all 5 credible variants to be located within the gene, or how was this done? Could the authors include information about that on page 8 and in the method section page 16.

Additionally, at page 8 - is it the 597 loci with less than 5 credible variants from the trans-ancestry analysis that are used as basis for mapping genes that goes into the enrichment analysis? I assume it is, but it was not clear to me.

4) When intersecting the 597 trans-ancestry loci with their TWAS results, the authors write that they "cross-reference" loci with < 5 credible variants with their TWAS results and end up with 52 genes of interest. Also here, it is not clear to me how this "cross-reference" was done – did the authors require all five credible variants to be located within the gene? Could the authors please provide this information in the method section.

5) I would prefer a summary section without the part trying to explain the cross-ancestry PRS results by the omnigenic model. Even though it might be okay powered for PRS analyses The Add Health target sample is not that big after all, and I think it is too early to make any strong conclusions. Thus, the section becomes very speculative, and I think the manuscript would be stronger by just focusing on the well-supported results/conclusions.

Referee #2 (Remarks to the Author):

The authors have performed a considerable amount of work to address my previous comments, and I commend them for their effort. I have no new comments.

Referee #3 (Remarks to the Author):

The authors have made comprehensive changes in the manuscript in line with the suggestions made by myself and other reviewers. I have no further suggestions for improvement, great work!

Referee #4 (Remarks to the Author):

I'm happy with the revisions made in response to my previous comments. Some further minor comments:

L417 you use R^2 for the LD coefficient, I am more used to r^2 for that with R^2 used for variance explained - but no problem I think the usage is clear.

L613 "finding sign concordance estimates of 63.4–80%, all of which were significantly higher than would be expected if our results were driven entirely population stratification or cryptic relatedness" you should state what the range is over. I won't insist on any change at this stage but it's a very weak claim because the null of no true effect (everything only due to confounding) is so implausible as to be of no interest, and meanwhile there could be a mix of true results and effect size inflation due to confounding, this analysis gives us no help on that question.

Author Rebuttals to First Revision:

Referees' comments:

Referee #1 (Remarks to the Author):

First, I would like to congratulate the authors with their very interesting and comprehensive work. There are many new and exciting findings which include loci discovery, effect size modulation by ancestry, improved fine-mapping when including ancestry information and careful evaluation of PRS prediction across ancestries. In my opinion, the study merits publication in Nature.

The authors have cleared most of my points raised. I only have a few minor comments:

1) I have an additional comment regarding the DrnkWk phenotype. In the current study the authors find a non-significant small positive genetic correlation of DrnkWk with college degree ($r_g = 0.04$, $P = 0.03$) and job codes related to higher education (reflecting other studies finding alcohol consumption to have a positive genetic correlation with educational attainment (Kranzler et al. 2019)). Alcohol use disorder is known to demonstrate, a significant negative genetic correlation with educational attainment. So, I would be cautious stating that DrnkWk “measures” alcohol use disorder because there is a part of the genetic component underlying DrnkWk that differs from that underlying AUD (which the authors also highlight). I would therefore suggest a more moderate wording simply stating that there exist a genetic overlap, e.g., : “DrnkWk was most strongly correlated with problematic drinking behaviors ($r_g = 0.52-0.70$), suggesting an overlap in the genetic architecture of DrnkWk and measures of alcohol use problems”.

Thank you for raising this point. We agree that there is at least some genetic component underlying DrnkWk that differs from that which underlies alcohol use disorder. We have adopted the suggested, more moderate, wording on page 11 of the main text.

2) Please also state the P-value threshold for claiming significant genetic correlation, I could not find that in the text.

We used Bonferroni corrected P-value threshold for 1,141 UK Biobanks phenotypes (i.e., $P \leq 4.38 \times 10^{-5}$) for claiming a genetic correlation significantly different from 0. We have more clearly stated this in the methods section on heritability and genetic correlations.

3) I am very happy to see that the authors have included enrichment analyses of high priority genes. Thank you very much! I have a few questions related to this. On page 8 the authors write: “To characterize genes prioritized from fine-mapping, we conducted a series of functional...”. It is not clear how genes are linked to credible variants – do the authors require all 5 credible variants to be located within the gene, or how was this done? Could the authors include information about that on page 8 and in the method section page 16.

We did not require that all 5 variants in the credible interval be located within a gene. Rather, we used Ensembl to map each of the variants in a credible interval fine-mapped to less than five variants to a gene based on the hg38 UCSC knownGene annotation database. This results in a total of 583 unique genes across all phenotypes with some credible intervals containing more than one gene. Approximately 58% of the variants were located within a gene. The remainder were located in intergenic regions (mean distance from the nearest gene of 181kb with a standard deviation of 303kb). For these intergenic variants we assigned them to the closest gene regardless of whether it was upstream or downstream. This was done, in part, based on prior feedback that the best predictor for the causal gene is still distance from the index variant to the gene. We have added a column in Supplementary Table 3 that flags whether or not a given variant is located within the assigned gene. The ‘high priority’ gene list includes all 583 genes whether or not they were mapped from variants within, or outside of, the gene. We have added detail about this procedure on page 8 of the main text, in the methods section, and on page 17 of the supplementary text.

Additionally, at page 8 - is it the 597 loci with less than 5 credible variants from the trans-ancestry analysis that are used as basis for mapping genes that goes into the enrichment analysis? I assume it is, but it was not clear to me.

Yes, the 597 loci with fewer than 5 variants in the credible intervals from the trans-ancestry results formed the basis of the fine-mapped enrichment analyses. We have added a note of clarification about this on page 8 of the main text.

4) When intersecting the 597 trans-ancestry loci with their TWAS results, the authors write that they “cross-reference” loci with < 5 credible variants with their TWAS results and end up with 52 genes of interest. Also here, it is not clear to me how this “cross-reference” was done – did the authors require all five credible variants to be located within the gene? Could the authors please provide this information in the method section.

To identify the 52 genes of interest, we intersected the list of variants from credible intervals containing less than five variants with the cis-eQTLs from the trans-ancestry TWAS. So, in this case, the intersection was performed at the variant level. For example, if variant A is in a well fine-mapped interval and is a cis-eQTL for a gene that is significantly associated in the trans-ancestry TWAS results, we would select that gene as one of interest. In this way, the variants were not required to be located within a gene but were required to be cis-eQTLs for a given gene. We have changed wording on page 10 of the main text to clarify how we identified the list of genes of interest.

5) I would prefer a summary section without the part trying to explain the cross-ancestry PRS results by the omnigenic model. Even though it might be okay powered for PRS analyses The Add Health target sample is not that big after all, and I think it is too early to make any strong conclusions. Thus, the section becomes very speculative, and I think the manuscript would be stronger by just focusing on the well-supported results/conclusions.

Thank you for raising this issue. We have removed discussion of the omnigenic model in the summary section of the main text and replaced it with a simple assertion that explanations for the discrepancy between variant heterogeneity and polygenic transferability have been forwarded, with a cautionary note that more stringent research will be necessary to make broad claims about heredity.

Referee #2 (Remarks to the Author):

The authors have performed a considerable amount of work to address my previous comments, and I commend them for their effort. I have no new comments.

Thank you!

Referee #3 (Remarks to the Author):

The authors have made comprehensive changes in the manuscript in line with the suggestions made by myself and other reviewers. I have no further suggestions for improvement, great work!

Thank you!

Referee #4 (Remarks to the Author):

I'm happy with the revisions made in response to my previous comments. Some further minor comments:

L417 you use R^2 for the LD coefficient, I am more used to r^2 for that with R^2 used for variance explained - but no problem I think the usage is clear.

Thank you for catching this! It has been corrected to r^2 .

L613 "finding sign concordance estimates of 63.4–80%, all of which were significantly higher than would be expected if our results were driven entirely population stratification or cryptic relatedness" you should state what the range is over. I won't insist on any change at this stage but it's a very weak claim because the null of no true effect (everything only due to confounding) is so implausible as to be of no interest, and meanwhile there could be a mix of true results and effect size inflation due to confounding, this analysis gives us no help on that question.

We have added "across phenotypes" to page 15 to clarify what the range of estimates refers to. We agree that the null hypothesis that all effects are due entirely to confounding is relatively weak. Our intent was to combine these sign concordance tests with other approaches (i.e., the RATES replicability analyses, LD score regression, and sign concordance with 23andMe) to assess, to the extent possible, the effects of confounding and replicability of our results. We have added 95% confidence intervals around all sign concordance estimates on page 15 of the

supplementary text and have noted these the observed concordance estimates are similar, or larger, in magnitude to those found in other large-scale association studies in the methods section.

Reviewer Reports on the Second Revision:

Referees' comments:

Referee #1 (Remarks to the Author):

Thank you very much for clarifying my questions, I have no further suggestions for improvement.

Well performed study with very interesting findings!